# MRVF: MULTI-ROUND VALUE FACTORIZATION WITH GUARANTEED ITERATIVE IMPROVEMENT FOR MULTI-AGENT REINFORCEMENT LEARNING

## ABSTRACT

Value factorization restricts the joint action value in a monotonic form to enable efficient search for its optimum. However, the representational limitation of monotonic forms often leads to suboptimal results in cases with highly non-monotonic payoff. Although recent approaches introduce additional conditions on factorization to address the representational limitation, we propose a novel theory for convergence analysis to reveal that single-round factorizations with elaborated conditions are still insufficient for global optimality. To address this issue, we propose a novel Multi-Round Value Factorization (MRVF) framework that refines solutions round by round and finally obtains the global optimum. To achieve this, we measure the non-negative incremental payoff of a solution relative to the preceding solution. This measurement enhances the monotonicity of the payoff and highlights solutions with higher payoff, enabling monotonic factorizations to identify them. We evaluate our method in three challenging environments: non-monotonic one-step games, predator-prey tasks, and StarCraft II Multi-Agent Challenge (SMAC). Experiment results demonstrate that our MRVF outperforms existing value factorization methods, particularly in scenarios highly non-monotonic payoff.

## 1 INTRODUCTION

Multi-agent reinforcement learning (MARL) effectively addresses many real-world problems, such as robotics (Hüttenrauch et al., 2019), automated warehouses (Tao et al., 2024), and games (Berner et al., 2019). MARL's core issue is finding the optimal action in an action space that grows exponentially with the number of agents. To address this issue, value factorization methods represent the joint action value in a specific form. Early value factorization methods, such as VDN (Sunehag et al., 2017) and QMIX (Rashid et al., 2020b), represent the joint action value through a monotonic mix of individual action values. This monotonic relationship ensures that the optimal joint action can be obtained by searching through individual action spaces. However, because of the monotonic particularity, these methods struggle to obtain the global optimum in non-monotonic cases.

To mitigate the representational limitation in VDN and QMIX, the following methods propose new fitting functions (e.g., QPLEX (Wang et al., 2021) and ResQ (Shen et al., 2022)) or loss functions (e.g., QTRAN (Son et al., 2019) and WQMIX (Rashid et al., 2020a)). These designs can be treated as conditions on the current greedy action to enhance its convergence to the optimal action. However, the effectiveness of these conditions relies on a strong assumption: the current greedy action is the optimal action, which means that the optimal action has already been found. We find that when the current greedy action is among certain suboptimal actions, the greedy action may converge to such suboptimal actions, resulting in poor performance of these methods. **One of our contributions is that** we introduce a theoretical tool with a novel concept, **stable point**, to describe the convergence of greedy action under value factorization. Using this tool, we explain how existing methods converge to suboptimal solutions and provide specific cases where they fail to obtain the optimum.

Theoretically, we conclude that no matter how elaborate the conditions on greedy action are, monotonic factorization is almost impossible to obtain the optimal solution for non-monotonic cases. However, if a non-monotonic payoff is transformed into a monotonic one, we can easily obtain the optimal solution with monotonic factorization. To achieve this, **another contribution of ours is**

**that** we propose a multi-round value factorization (MRVF) framework. As illustrated in Figure 1, in each round, we replace the payoff with the payoff increment (non-negative), the payoff clipped by that of the preceding solution. This process gradually transforms the original payoff into a monotonic one round by round, thereby enabling monotonic factorization to achieve the optimal solution. Furthermore, using the theory of stable points, we prove that the solution obtained based on the increment is strictly improved from the previous round. This guarantees that the optimal solution can be obtained with a sufficient number of iterations.

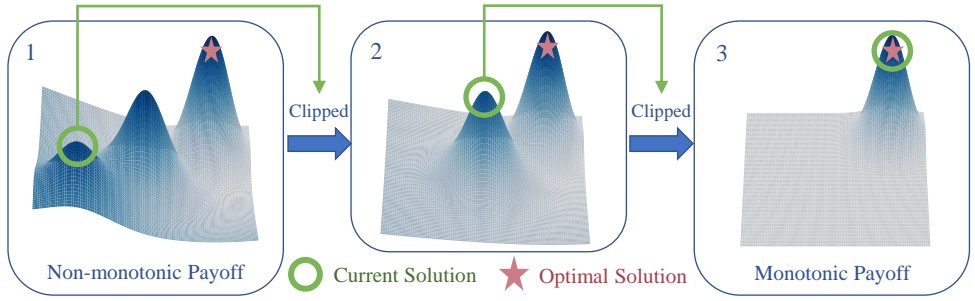

Figure 1: The process of transforming a non-monotonic payoff into a monotonic one round by round. At the beginning, MRVF obtains a suboptimal solution for the non-monotonic payoff. However, by clipping off these suboptimal values, we transform the payoff into a monotonic one that only leaves optimal values, thereby enabling monotonic factorization to find the optimal solution. The definition of monotonic payoff is presented in Appendix B.1.

To illustrate the superiority of MRVF, we conduct experiments on randomly generated one-step games that require significant coordination to get bonus, or otherwise a penalty. In addition, we evaluate MRVF on more challenging tasks such as the predator-prey task (Böhmer et al., 2020) and the StarCraft II MARL tasks (Whiteson et al., 2019). The experimental results show that MRVF outperforms existing value factorization methods, particularly in challenging scenarios that require high levels of agent coordination. The ablation study demonstrates the importance of strictly improving current solutions by using the payoff increment as the target value.

## 2 BACKGROUND

### 2.1 DEC-POMDP

In cooperative multi-agent systems, agents interact with the environment to achieve common objectives. This process can be modeled as a decentralized partially observable Markov decision process (Dec-POMDP) (Oroojlooy & Hajinezhad, 2023), defined by a tuple $< \mathcal{S}, \mathcal{U}, P, O, R, \gamma, n >$, in which $n$ is the number of agents, $\mathcal{S}$ is the state space, $\mathcal{U} = \mathcal{U}_1 \times \mathcal{U}_2 \times \cdots \times \mathcal{U}_n$ is the action space, $P(\boldsymbol{s}'|\boldsymbol{s}, \boldsymbol{u}) : \mathcal{S} \times \mathcal{U} \times \mathcal{S} \rightarrow [0, 1]$ is the transition probability between the states, $O : \mathcal{S} \rightarrow \mathcal{O}$ is the joint observation function where $\mathcal{O}$ is the joint observation space, $r \sim R : \mathcal{S} \times \mathcal{U} \rightarrow \mathbb{R}$ is the reward, $\gamma \in (0, 1]$ is the discount, agent have an action-observation history $\boldsymbol{\tau} \in \mathcal{T} \equiv (\mathcal{O} \times \mathcal{U})^*$. Similarly to RL with a single agent, the objective of MARL is to find the policy $\boldsymbol{\pi} = (\pi_1, \pi_2, \cdots, \pi_n) : \mathcal{T} \times \mathcal{U} \rightarrow [0, 1]$ that maximizes the joint action value $Q_{\mathrm{jt}}(\boldsymbol{s}, \boldsymbol{u}) = \mathrm{E}_{\boldsymbol{\pi}}[\sum_t \gamma^t r_t | \boldsymbol{s}, \boldsymbol{u}]$, the expectation of return. However, obtaining the global optimal action $\boldsymbol{u}^* = \arg\max_{\boldsymbol{u}} Q_{\mathrm{jt}}(\boldsymbol{s}, \boldsymbol{u})$ by searching the large action space is intractable for value-based learning.

### 2.2 MONOTONIC VALUE FACTORIZATION

Value factorization methods approximate the joint action value $Q_{\mathrm{jt}}$ with a specifically formed $Q_{\mathrm{tot}}$ by minimizing $L_{\mathrm{tot}}$, the Mean Squared Error (MSE) between $Q_{\mathrm{jt}}$ and $Q_{\mathrm{tot}}$. For example, VDN (Sunehag et al., 2017) factorizes $Q_{\mathrm{tot}}$ into the sum of individual Qs: $Q_{\mathrm{tot}} = \sum_i Q_i$. And QMIX (Rashid et al., 2020b) uses a monotonic function $f_{\mathrm{mon}}$ to represent the relationship: $Q_{\mathrm{tot}} = f_{\mathrm{mon}}(Q_1, Q_2, \cdots, Q_n)$ where $\frac{\partial f_{\mathrm{mon}}}{\partial Q_i} \geq 0, \forall i \in \{1, 2, \cdots, N\}$. The monotonicity of $Q_{\mathrm{tot}}$ with respect to $Q_i$ ensures the alignment of their optimal actions, which is the Individual-Global-

Max (IGM) principle, defined as follows [1]:

$$\prod_{i=0}^{n} \arg\max_{u_i} Q_i(\tau_i, u_i) \subseteq \arg\max_{\boldsymbol{u}} Q_{\text{tot}}(\boldsymbol{\tau}, \boldsymbol{u}) \qquad (1)$$

where $\prod_{i=0}^{n} \arg\max_{u_i} Q_i(\tau_i, u_i) = \arg\max_{u_1} Q_1(\tau_1, u_1) \times \cdots \times \arg\max_{u_n} Q_n(\tau_n, u_n)$, and $Q_i : \mathcal{T}_i \times \mathcal{U}_i \to \mathbb{R}$ is the individual action value of agent $i$. Here we define $\overline{\boldsymbol{u}} \in \prod_{i=0}^{n} \arg\max_{u_i} Q_i(\tau_i, u_i)$ as the greedy action. If the IGM principle is satisfied, and $Q_{\text{tot}}$ approximate $Q_{\text{jt}}$ precisely, it is easy to get the maximum of $Q_{\text{jt}}$ by searching that of $Q_i$.

## 2.3 GRADIENT-FREE COMPONENTS IN VALUE FACTORIZATION

Value factorization methods use gradient descent to update gradient-based components, such as individual Q values and the mixing network $f_{\text{mon}}$. However, not all components in the factorization are gradient-based. Specifically, the components taking greedy action as a parameter are gradient-free. To distinguish the meanings of $\overline{\boldsymbol{u}}$ as the parameter and greedy action, we refer to $\widetilde{\boldsymbol{u}}$ as the parameter of these components, where $\overline{\boldsymbol{u}}$ is assigned to $\widetilde{\boldsymbol{u}}$. Gradient-free components are common in value factorization, for example, the decentralized $\epsilon$-greedy policy $\boldsymbol{\pi}$

$$\boldsymbol{\pi}(\boldsymbol{u}|\boldsymbol{\tau}) = (\frac{\epsilon}{|\mathcal{U}|})^{n-m}(1 - \epsilon + \frac{\epsilon}{|\mathcal{U}|})^m \qquad (2)$$

where $m = |\{i|\boldsymbol{u}_i = \widetilde{\boldsymbol{u}}_i\}|$. Recent methods introduce additional terms involving $\widetilde{\boldsymbol{u}}$ into monotonic factorization. For example, the weight $w$ of the weighted $L_{\text{tot}}$ in WQMIX (Rashid et al., 2020a) [2]

$$w(\boldsymbol{\tau}, \boldsymbol{u}) = \begin{cases} 1 & \hat{Q}_{\text{jt}}(\boldsymbol{\tau}, \boldsymbol{u}) > \hat{Q}_{\text{jt}}(\boldsymbol{\tau}, \widetilde{\boldsymbol{u}}) \ or \ \boldsymbol{u} = \widetilde{\boldsymbol{u}} \\ \alpha < 1 & otherwise \end{cases} \qquad (3)$$

where $\hat{Q}_{\text{jt}}$ is an approximation of $Q_{\text{jt}}$, and the residual mask $w_{\text{r}}$ in ResQ (Shen et al., 2022)

$$w_{\text{r}}(\boldsymbol{\tau}, \boldsymbol{u}) = \begin{cases} 0 & \boldsymbol{u} = \widetilde{\boldsymbol{u}} \\ 1 & otherwise \end{cases} \qquad (4)$$

## 3 SUBOPTIMALITY OF EXISTING SINGLE-ROUND FACTORIZATION

In this section, we discuss why the policy (or the greedy action) is trapped in local optima during training, which results in existing value factorization methods obtain a suboptimal result. To explain this, we need to analyze the convergence of greedy actions in value factorization. For the convergence analysis, existing works only analyze the cases when $\widetilde{\boldsymbol{u}}$ (a parameter of gradient-free components) is the optimal action, which is insufficient. In contrast, we provide an analytical approach to determine what outcomes that $\overline{\boldsymbol{u}}$ (the greedy action) converges to for general cases of $\widetilde{\boldsymbol{u}}$, and reveal situations where $\overline{\boldsymbol{u}}$ converges to suboptimal actions under certain $\widetilde{\boldsymbol{u}}$.

### 3.1 TRANSITION OF THE GREEDY ACTION

To introduce the analysis of the greedy action's convergence, we first illustrate how the greedy action changes during training. We provide an example in Figure 2, where we observe that the transition of greedy actions is similar to that of a Markov process: $\widetilde{\boldsymbol{u}}$ acts as the current "state" (controlling the transitions) and $\overline{\boldsymbol{u}}$ as the next "state". This process emerges because updates to gradient-free parameters (indicated by $\widetilde{\boldsymbol{u}}$) and gradient-based parameters (indicated by $\overline{\boldsymbol{u}}$) are asynchronous, with their relationship shown in Figure 3.

Combining Figure 3, we illustrate the process in Figure 2 as follows:

(1) Initially, $\widetilde{\boldsymbol{u}} = (1, 3)$ (top right) generates an all-ones weight matrix.

---
[1]We use "$\subseteq$" instead of "$=$" in Equation (1), as $Q_{\text{tot}}$ may have more maxima than individual $Q$s (Table 1).
[2]It stands for Centrally-Weighted QMIX.

(2) Minimizing $L_{\text{tot}} = w * (Q_{\text{tot}} - Q_{\text{jt}})^2$ under the all-ones weight $w$ yields the $Q_{\text{tot}}$ (left table), corresponding to $\overline{u} = (3, 3)$ (bottom right).

(3) $\widetilde{u}$ is assigned the new value $\overline{u} = (3, 3)$, thereby altering the weight matrix (shaded regions).

(4) Minimizing $L_{\text{tot}}$ with the weight under $\widetilde{u} = (3, 3)$ produces a new $Q_{\text{tot}}$ (middle table).

(5) $\overline{u}$ keeps on changing (steps 2-4) until $\overline{u} = \widetilde{u}$ is reached (right table).

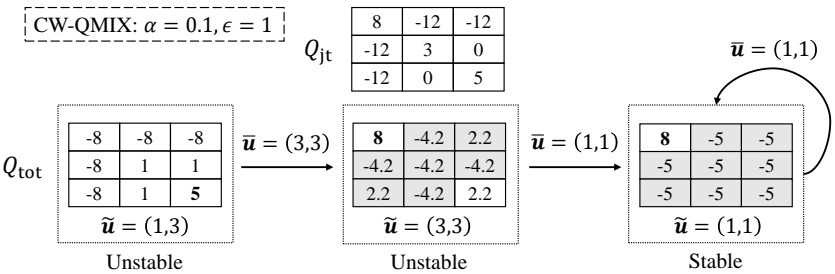

Figure 2: The transition of greedy action in WQMIX with $\alpha = 0.1$ under uniform visitation ($\epsilon = 1$). This process begins with $\widetilde{u} = (1, 3)$ (top right) and stabilizes at $\overline{u} = (1, 1)$ (top left). The cells in $Q_{\text{tot}}$ where $w(s, u) = \alpha$ are shaded, and the value corresponding to $\overline{u} = (row, column)$ is in bold.

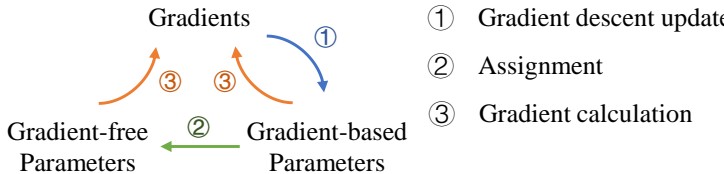

Figure 3: The relationship between gradient-based and gradient-free parameters. The gradient-free parameters are updated according the values of gradient-based parameters rather than the gradients, leading to asynchronous updates between the two parameters.

## 3.2 CONVERGENCE OF THE GREEDY ACTION

From the discussion above, the transition of $\overline{u}$ typically begins in a transient phase (steps 2-4) and ends in a steady phase (step 5). The steady phase of this transition is what we are interested in, which reflects the convergence result of greedy actions. Based on the above discussion, **we use stable points to describe the convergence of $\overline{u}$**, defined as follows:

**Definition 1** (Stable Point). *In a specific value factorization framework, for a joint action value $Q_{\text{jt}}$ and state $s \in \mathcal{S}$, if $\check{u}$ is a stable point of $Q_{\text{jt}}(s, \cdot)$, then for $\widetilde{u} = \check{u}$ in the optimization of minimizing $L_{tot}$, the converged $\overline{u}$ satisfies $\overline{u} = \check{u}$.*

Note that multiple stable points may exist in value factorization, any of which could be the final result of greedy actions. To analyze the possible outcomes of greedy actions, we need to determine whether an action is a stable point, which is described as follows:

(1) Set $\widetilde{u} = \check{u}$ to generate the gradient-free parameters.

(2) Minimize $L_{\text{tot}}$ under $\widetilde{u} = \check{u}$ to obtain $Q_{\text{tot}}$ (multiple solutions may exist) and corresponding $\overline{u}$.

(3) If any $\overline{u}$ satisfies $\overline{u} = \check{u}$, then $\check{u}$ is a stable point.

(4) Furthermore, if $\overline{u} = \check{u}$ holds for all solutions, $\check{u}$ is a strongly stable point. Otherwise, $\check{u}$ is a weakly stable point.

The difference between mixer types lies in Step 2: For VDN-style mixers, we obtain $Q_{\text{tot}}$ using gradient descent. And for QMIX-style mixers, we obtain $Q_{\text{tot}}$ based on the Ideal QMIX assumption: The Ideal QMIX yields the optimal $Q_{\text{tot}}$ that minimizes $L_{\text{tot}}$ under any given $\widetilde{u}$ (Appendix B.2).

### 3.3 SUBOPTIMALITY OF EXISTING FACTORIZATION

We analyze stable points in existing methods to demonstrate cases where greedy actions converge to suboptimal actions, which usually happens in environments with highly non-monotonic payoff. Then, we derive the conditions under which the greedy action converges solely to the optimal action.

**WQMIX:** WQMIX can not guarantee the convergence to the optimal action. In some cases, as shown in Table 1, the optimal action is not a stable point for any $\alpha$, which means that the greedy action does not converge to the optimal action.

|   |   |    |
|---|---|----|
| 4 | 0 | -8 |
| 0 | 3 | 0  |
| -8| 0 | -8 |

(a) $Q_{\mathrm{jt}}$

|          |          |          |
|----------|----------|----------|
| 1        | $\alpha$ | $\alpha$ |
| $\alpha$ | $\alpha$ | $\alpha$ |
| $\alpha$ | $\alpha$ | $\alpha$ |

(b) $w$

$L_{\mathrm{tot}}$=70$\alpha$

|     |    |    |
|-----|----|----|
| **4** | 1  | -4 |
| 1   | 1  | -4 |
| -4  | -4 | -8 |

(c) $Q_{\mathrm{tot}}^*$ for $\overline{u} = u^*$

$L_{\mathrm{tot}}$=33$\alpha$

|    |     |    |
|----|-----|----|
| 4  | 4   | -8 |
| 4  | **4** | 0  |
| -8 | 0   | -8 |

(d) $Q_{\mathrm{tot}}$ with $\overline{u} \neq u^*$

Table 1: An example for WQMIX in which $u^*$ is not a stable point. (a): The matrix of $Q_{\mathrm{jt}}$ with the maximum value of 4. (b): The weight $w$ of WQMIX where $\widetilde{u} = u^*$. (c): The $Q_{\mathrm{tot}}$ with minimum $L_{\mathrm{tot}}$ under the condition that $\overline{u} = u^*$. (d): Another $Q_{\mathrm{tot}}$ where $\overline{u} \neq u^*$ achieves less $L_{\mathrm{tot}}$ than (c). Therefore, $u^*$ is not a stable point in this case.

**QPLEX:** QPLEX proves that if $\widetilde{u}$ happens to be the optimal action, then $\overline{u}$ can converge to the optimal action. However, we find that multiple suboptimal stable points exist in QPLEX, which means that if $\widetilde{u}$ is certain suboptimal actions, $\overline{u}$ will converge to them (examples in Appendix D).

**ResQ:** ResQ proves that $\overline{u}$ can converge to the optimal action when $\widetilde{u}$ is the optimal action (similar to QPLEX). However, we find that although the optimal action is the only stable point, it is always a weak stable point (proof in Appendix C.2). This means that even if the optimal action has been found ($\widetilde{u} = u^*$), it may be lost subsequently.

From the above discussion, we conclude that in single-round value factorization, if the greedy action is to be guaranteed to converge to the optimal action, then the optimal action must be the unique strongly stable point, which means:

(1) The optimal action must be a strong stable point (WQMIX and ResQ fail)

(2) All suboptimal actions must be unstable points (QPLEX fails).

## 4 APPROACH OPTIMUM THROUGH MULTI-ROUND FACTORIZATION

In Section 3.3, we provide the condition under which the greedy action converges exclusively to the optimal action. This condition is highly stringent, to the extent that existing single-round factorization methods struggle to satisfy them. Therefore, we propose a multi-round factorization framework with a lenient condition for achieving optimality: **the greedy action must be strictly improved round by round (Strict Improvement Condition).** Satisfying this condition guarantees that the greedy action in a certain round is the optimal one (Theorem 1).

**Theorem 1.** *For a joint action value $Q_{\mathrm{jt}}$ with a finite action space $\mathcal{U}$ and $s \in \mathcal{S}$, consider a sequence $\{\overline{u}^k | \overline{u}^k \in \mathcal{U}\}$, if $\forall k > 1, Q_{\mathrm{jt}}(s, \overline{u}^k) > Q_{\mathrm{jt}}(s, \overline{u}^{k-1})$ when $\overline{u}^{k-1} \neq u^*$, then $\exists K > 0, \overline{u}^K = u^*$. (Proof in Appendix C.1)*

### 4.1 DESIGNS FOR THE STRICT IMPROVEMENT CONDITION

The key point in MRVF is designing the backward computation of $Q_{\mathrm{tot}}$. Typically, $\hat{Q}_{\mathrm{jt}}$, a payoff measurement, serves as the target value of $Q_{\mathrm{tot}}$. In this paper, we replace it with the action-value increment for round $k > 1$. As shown in Equation (5), this increment is computed by clipping the original $\hat{Q}_{\mathrm{jt}}$ with its value at the preceding greedy action $\overline{u}_t^{k-1}$, with negative values set to zero.

$$L_{\mathrm{tot}} = \mathrm{E}_{\boldsymbol{\tau}_t, \boldsymbol{u}_t^k \sim \boldsymbol{\pi}_k}[(Q_{\mathrm{tot}}(\boldsymbol{\tau}_t, \overline{\boldsymbol{u}}_t^{k-1}, \boldsymbol{u}_t^k) - \max\{\hat{Q}_{\mathrm{jt}}(\boldsymbol{\tau}_t, \boldsymbol{u}_t^k) - \hat{Q}_{\mathrm{jt}}(\boldsymbol{\tau}_t, \overline{\boldsymbol{u}}_t^{k-1}), 0\})^2], \quad k > 1 \ (5)$$

where $\max\{\hat{Q}_{\mathrm{jt}}(\boldsymbol{\tau}_t, \boldsymbol{u}_t^k) - \hat{Q}_{\mathrm{jt}}(\boldsymbol{\tau}_t, \overline{\boldsymbol{u}}_t^{k-1}), 0\}$ is the action-value increment.

Intuitively, the action-value increment enhances the monotonicity of $\hat{Q}_{\mathrm{jt}}$ round by round, thereby enabling monotonic factorization to find the optimal solution easily. **Theoretically, it ensures the strict improvement of greedy action when $\overline{\boldsymbol{u}}_t^{k-1} \neq \boldsymbol{u}^*$.** This is grounded in a property of monotonic factorization: the greedy action under QMIX will not converge to actions with the lowest target value of $Q_{\mathrm{tot}}$, since these lowest-value actions are not stable points (Theorem 2). Therefore, with the the action-value increment, $\overline{\boldsymbol{u}}_t^k$ only converge to actions that are superior to $\overline{\boldsymbol{u}}_t^{k-1}$, since all $\boldsymbol{u}_t^k$ with $\hat{Q}_{\mathrm{jt}}(\boldsymbol{\tau}_t, \boldsymbol{u}_t^k) \leq \hat{Q}_{\mathrm{jt}}(\boldsymbol{\tau}_t, \overline{\boldsymbol{u}}_t^{k-1})$ are assigned the minimum value, zero.

**Theorem 2.** *Consider a non-constant joint action value $Q_{\mathrm{jt}}(\boldsymbol{s}, \cdot)$ ($\forall \boldsymbol{s} \in \mathcal{S}, \forall C \in \mathbb{R}, Q_{\mathrm{jt}}(\boldsymbol{s}, \cdot) \not\equiv C$) with finite state space $\mathcal{S}$ and action space $\mathcal{U}$. For the ideal QMIX defined in Equation (12) and the centralized $\epsilon$-greedy policy $\boldsymbol{\pi}$ defined in Equation (8), we have $\forall \boldsymbol{s} \in \mathcal{S}, \exists E \in (0, 1), \forall \epsilon \in (0, E), \forall \boldsymbol{u} \in \arg\min_{\boldsymbol{u}} Q_{\mathrm{jt}}(\boldsymbol{s}, \boldsymbol{u})$, $\boldsymbol{u}$ is not a stable point. (Proof in Appendix C.1)*

We illustrate in Table 2 how the target value of $Q_{\mathrm{tot}}$ shapes during the multi-round process. For the first round ($k = 1$), we use the original $\hat{Q}_{\mathrm{jt}}$ as the target for $Q_{\mathrm{tot}}$, as shown in Equation (6).

$$L_{\mathrm{tot}} = \mathrm{E}_{\boldsymbol{\tau}_t, \boldsymbol{u}_t^k \sim \boldsymbol{\pi}_k}[(Q_{\mathrm{tot}}(\boldsymbol{\tau}_t, \boldsymbol{u}_t^k) - \hat{Q}_{\mathrm{jt}}(\boldsymbol{\tau}_t, \boldsymbol{u}_t^k))^2], \quad k = 1 \tag{6}$$

| 8 | -12 | -12 |
|---|---|---|
| -12 | 3 | 0 |
| -12 | 0 | **5** |

(a) $k = 1$

| **3** | 0 | 0 |
|---|---|---|
| 0 | 0 | 0 |
| 0 | 0 | 0 |

(b) $k = 2$

| 0 | 0 | 0 |
|---|---|---|
| 0 | 0 | 0 |
| 0 | 0 | 0 |

(c) $k = 3$

Table 2: This table shows the target of $Q_{\mathrm{tot}}$ in three round, in which the value corresponding to $\overline{\boldsymbol{u}}$ is in bold when it is unique. (a): The target of $Q_{\mathrm{tot}}$ in the first round which is the original $\hat{Q}_{\mathrm{jt}}$. (b): The target of $Q_{\mathrm{tot}}$ in the second round which is clipped with $\hat{Q}_{\mathrm{jt}}(\boldsymbol{\tau}_t, \overline{\boldsymbol{u}}_t^1) = 5$. (c): The target of $Q_{\mathrm{tot}}$ in the final round where any $\boldsymbol{u}$ can be $\overline{\boldsymbol{u}}^3$.

The forward computation in step $t$ generates the final action $\boldsymbol{u}_t$ (or the greedy final action $\overline{\boldsymbol{u}}_t$ during evaluation) that actually interacts with the environment. Note that the final action is not always the action in the final round. As shown in Table 2, the greedy action in the round $k = 2$ is the optimal action. As a result, the greedy action can possibly converge to any action in round $k = 3$, which violates the Strict Improvement Condition. Therefore, early termination is necessary when the strict improvement is not achieved. Specifically, we check the strict improvement by comparing the $\hat{Q}_{\mathrm{jt}}$ values of the two greedy actions: When $\hat{Q}_{\mathrm{jt}}(\boldsymbol{\tau}_t, \overline{\boldsymbol{u}}_t^k) \leq \hat{Q}_{\mathrm{jt}}(\boldsymbol{\tau}_t, \overline{\boldsymbol{u}}_t^{k-1})$, the forward process in step $t$ is terminated with an output $\boldsymbol{u}_t^{k-1}$, otherwise the process proceed to the next round.

## 4.2 OVERVIEW OF THE ARCHITECTURE

The architecture of our methods is shown in Figure 4. The architecture consists of three crucial parts: individual Q networks, a mixing network with positive weights, and a joint action value network. Individual Q networks receive the action-observation history $\boldsymbol{\tau}_t$ and the greedy action of the preceding round $\overline{\boldsymbol{u}}_t^{k-1}$ (default action for the first round). Then, they output the actions of the current round $\boldsymbol{u}_t^k$ through the $\epsilon$-greedy action selector. The mixing network receives individual Q values and outputs $Q_{\mathrm{tot}}$. The joint action value network outputs $\hat{Q}_{\mathrm{jt}}$ to approximate the real $Q_{\mathrm{jt}}$. We update $\hat{Q}_{\mathrm{jt}}$ by minimizing the temporal difference (TD) error in Equation (7).

$$L_{\mathrm{jt}} = \mathrm{E}_{\boldsymbol{\tau}_t, \boldsymbol{u}_t \sim \boldsymbol{\pi}}[(\hat{Q}_{\mathrm{jt}}(\boldsymbol{\tau}_t, \boldsymbol{u}_t) - (r_t + \gamma \hat{Q}_{\mathrm{jt}}(\boldsymbol{\tau}_{t+1}, \overline{\boldsymbol{u}}_{t+1})))^2] \tag{7}$$

where $\overline{\boldsymbol{u}}_t$ is the greedy final action, and $\hat{Q}_{\mathrm{jt}}(\boldsymbol{\tau}_{t+1}, \overline{\boldsymbol{u}}_{t+1})$ approximates $\max_{\boldsymbol{u}_{t+1}} \hat{Q}_{\mathrm{jt}}(\boldsymbol{\tau}_{t+1}, \boldsymbol{u}_{t+1})$.

Within step $t$, both the forward and backward processes take $\boldsymbol{\tau}_t$ as input, and in each round, the individual Q networks are fed with $\boldsymbol{\tau}_t$ and $\overline{\boldsymbol{u}}_t^{k-1}$ to produce $Q_i$ and $\overline{\boldsymbol{u}}_t^k$. The difference is that the forward process computes and compares $\hat{Q}_{\mathrm{jt}}$ to determine whether to terminate early, while the backward process does not involve early termination and computes $\hat{Q}_{\mathrm{jt}}$ only once using the recorded $\boldsymbol{u}_t$ from the batch (not shown in Figure 4). In addition, $Q_{\mathrm{tot}}$ is computed only in the backward.

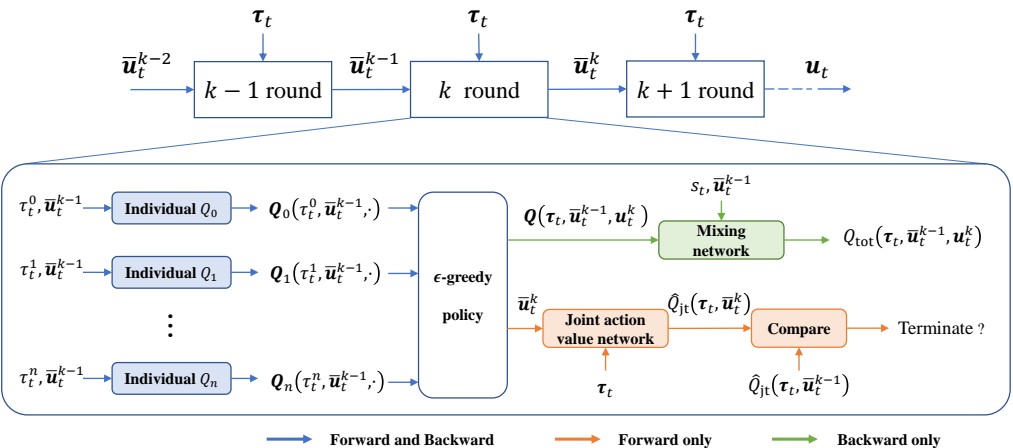

Figure 4: The architecture of multi-round value factorization framework.

## 4.3 SAMPLING

We approximate the expectation in $L_{\text{jt}}$ and $L_{\text{tot}}$ by sampling the action spaces. We design sampling strategies for multi-round frameworks to ensure stable training. For the final action $\boldsymbol{u}_t$ in $L_{\text{jt}}$, its sampling should not only cover the greedy final action and random actions, but also cover the greedy action in each round, which obtains an accurate target value in Equation (5). Therefore, during training, we select $\overline{\boldsymbol{u}}_t^k$) in a certain round $k$ as the final action with probability $p$. For $\boldsymbol{u}_t^k$ in $L_{\text{tot}}$, its distribution $\boldsymbol{\pi}_k$ is the centralized $\epsilon$-greedy policy defined in Equation (8) with $\widetilde{\boldsymbol{u}} = \overline{\boldsymbol{u}}^k$.

$$\boldsymbol{\pi}(\boldsymbol{u}|\boldsymbol{\tau}) = \begin{cases} 1 - \epsilon + \frac{\epsilon}{|\mathcal{U}|} & \boldsymbol{u} = \widetilde{\boldsymbol{u}} \\ \frac{\epsilon}{|\mathcal{U}|} & \boldsymbol{u} \neq \widetilde{\boldsymbol{u}} \end{cases} \tag{8}$$

The reason for not using the decentralized one (defined in Equation (2)) is that, according to Theorem 2, the centralized $\epsilon$-greedy policy is required to guarantee the Strict Improvement Condition. More importantly, since $Q_{\text{tot}}$ learns the actual payoff from $\hat{Q}_{\text{jt}}$, the centralized $\epsilon$-greedy policy allows the same actions to be sampled in $L_{\text{jt}}$ and $L_{\text{tot}}$ when sampling random actions, which prevents $Q_{\text{tot}}$ from learning regions where $\hat{Q}_{\text{jt}}$ exhibits underfitting.

## 5 EXPERIMENT

In this section, we evaluate the performance in one-step games, the predator-prey task (Böhmer et al., 2020) with increasing punishment, and a variety of SMAC (Whiteson et al., 2019) scenarios, among which one-step games and predator-prey tasks with large punishment are environments with highly non-montonic payoff (defined in Appendix B.1). We report the median performance, with the shaded area denoting the 25%-75% percentile (0%-100% percentile for one-step games). Our experimental setup is provided in Appendix E, and ablation results are presented in Appendix F.

## 5.1 ONE-STEP GAME

From the discussion in Section 3.3, we find that existing methods fail to achieve optimality when, at the global optimum, any deviation by an agent leads to significant negative rewards. To demonstrate this limitation of existing methods, we evaluate their performance in the *risk-reward* game, where agents must reach a consensus to obtain rewards, or otherwise they incur punishments, and higher rewards for consensus correspond to harsher punishments for dissension. These games are randomly generated by averaging individual rewards (positive if consensus is reached, negative otherwise), and more details are provided in the Appendix E.1. The *risk-reward* game is an extension of the matrix game (Son et al., 2019) (Shen et al., 2022) that allows the participation of more than two agents.

As shown in Figure 5, in cases with 3 agents and 5 actions, monotonic factorizations (QMIX (Rashid et al., 2020b) and NA$^2$Q (Liu et al., 2023)) obtain the smallest positive return, nearly zero, while

QTRAN (Wang et al., 2021) and WQMIX (Rashid et al., 2020a) only achieve a moderate positive return. In contrast, our method consistently attains the optimal return. When the number of agents increases to 5, the risk of obtaining positive returns increases, since consensus requires coordination among more agents. Existing methods rarely obtain positive returns, while our method still obtains the optimum with high probability. However, in cases with 5 agents and 8 actions, the positive rewards become too sparse to obtain. Despite this challenge, our method still obtains better results than others.

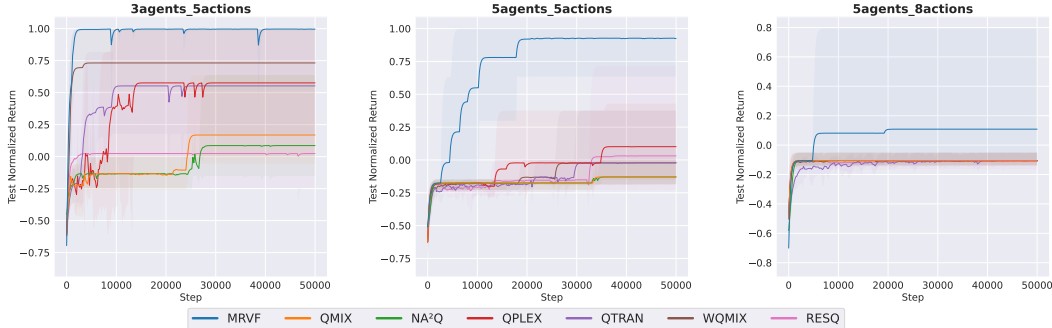

Figure 5: Test normalized return in the risk-reward games. The positive returns are normalized to $[0, 1]$ (0 corresponds to the smallest positive return, and 1 corresponds to the largest). The negative returns are normalized to $[-1, 0)$. Five random cases for each setting.

## 5.2 PREDATOR PREY

We conduct experiments on the predator-prey task involving 8 agents, where capturing a prey requires cooperation between at least two agents. Agents are punished for capturing the prey alone. We vary the punishment from 0 to -5 to evaluate its impact on performance. The non-monotonic property of the payoff increases with the punishment, since agents must act more cautiously when choosing the "capture" action to ensure consensus and avoid punishments.

The results in Figure 6 show that all methods perform well when the punishment is zero. However, as the punishment intensifies, existing methods avoid the "capture" action to prevent punishments, resulting in $return \approx 0$. Among baseline approaches, WQMIX performs adequately only under moderate punishments, while the performance of ResQ shows significant fluctuations due to its weak stability. In contrast, our method consistently achieves the best performance, even under the strongest punishment of -5. Combined with the results in *risk-reward* games, we conclude that our method outperforms existing methods in environments with highly non-monotonic payoff.

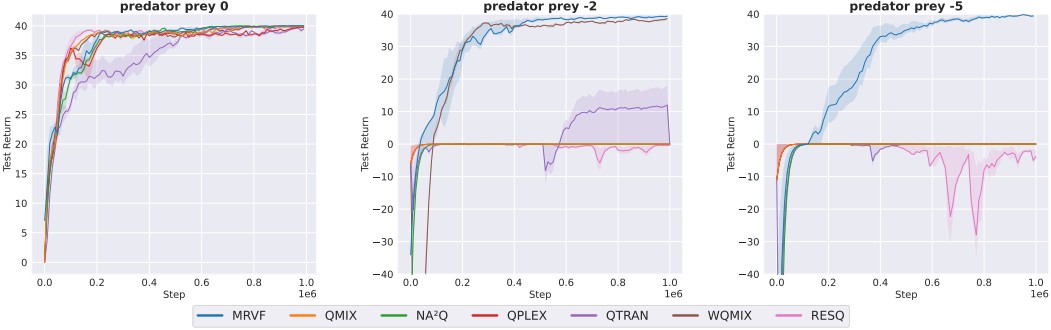

Figure 6: Test return in the predator prey tasks with punishments 0 (left), -2 (middle), and -5 (right). The non-monotonicity of the payoff increases with the punishment.

## 5.3 STARCRAFT II MULTI-AGENT CHALLENGE

We conduct experiments on the SMAC benchmark in scenarios of varying difficulty. Monotonic value factorization methods, including QMIX (Rashid et al., 2020b) and NA²Q (Liu et al., 2023),

are particularly well-suited for the SMAC because changes in an individual agent's action within a single step have little impact on overall performance (see Appendix E.3 for details). Nevertheless, as shown in Figure 7, MRVF even achieves an advantage over monotonic value factorization methods, particularly in *3s_vs_5z*, *2c_vs_64zg* and *bane_vs_bane* scenarios. Meanwhile, the final performance of MRVF is the best in most scenarios, as shown in Table 3. Combined with the results in the *risk-reward* games and the predator prey task, we conclude that MRVF performs the best in both monotonic and non-monotonic scenarios.

In contrast, other methods (QPLEX (Wang et al., 2021), ResQ (Shen et al., 2022)), QTRAN (Son et al., 2019) and WQMIX (Rashid et al., 2020a)) often converge to suboptimal stable points. This leads to a high variance in performance within a scenario and across scenarios. Within the same scenario, although they occasionally achieve strong performance (as seen in the upper bounds of the shaded regions in Figure 7), their average performance lags behind. In addition, across different scenarios, their performance may vary drastically. For example, WQMIX performs well in *bane_vs_bane* but poorly in both *5m_vs_6m* and *3s_vs_5z*. Therefore, MRVF achieves significant improvements in robustness and performance over them in almost all scenarios.

| Methods | **MRVF** | QMIX | NA$^2$Q | QPLEX | QTRAN | WQMIX | RESQ |
|---------|----------|------|---------|-------|-------|-------|------|
| Best | **6** | 3 | 4 | 1 | 1 | 1 | 3 |
| Worst | **0** | 1 | 0 | 3 | 5 | 2 | 1 |

Table 3: The number of SMAC scenarios where each method performs the best and the worst. We evaluate the final performance (averaged over steps after 1.5M) with a tolerance of $\pm 0.05$.

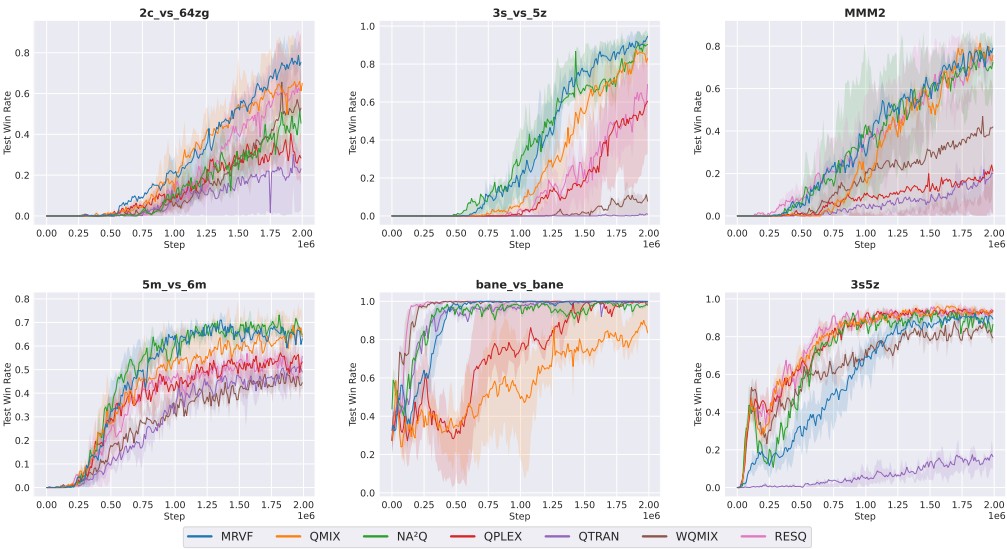

Figure 7: Test win rate in the SMAC benchmarks.

## 6 CONCLUSION

In this paper, we introduce a novel theoretical tool for studying the convergence of greedy action under value factorization, and propose MRVF, a novel framework for cooperative multi-agent reinforcement learning. In Section 3, we use this theoretical tool to derive the condition for global optimality in single-round factorization, and provide examples to demonstrate why existing methods struggle to satisfy this condition. In Section 4, we propose the condition for global optimality in multi-round factorization, which is strictly improving the greedy action round by round. To satisfy this condition, we design the forward and backward computation of MRVF. In addition, we design new sampling strategies suitable for multi-round factorization to ensure training stability. Experiments on non-monotonic one-step games, predator-prey tasks, and the StarCraft II Multi-Agent Challenge show the superior performance and robustness of MRVF compared to state-of-the-art value factorization methods.

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

# A  RELATED WORK

Reinforcement learning has been extensively studied in single-agent scenarios. However, when extended to multi-agent scenarios, a key issue is that the action space grows exponentially with the number of agents. To address this issue, existing approaches can be broadly categorized into two paradigms. The first paradigm treats other agents as part of the environment and focuses on learning individual value functions or policies. The second paradigm considers all agents as a unified entity and learns joint value functions with specific forms.

## A.1  OVERVIEW OF EXISTING WORK

**Independent Learning Methods**  Among the first category of methods, IQL (Tan, 1993) is one of the earliest methods where agents independently learn individual action values. However, this method does not account for the dynamic policies of other agents. Subsequent work addresses this limitation by learning individual policies with joint action values. For example, COMA (Foerster et al., 2018) introduces a counterfactual baseline in actor-critic framework, while MAPPO (Yu et al., 2022) adapts PPO (Schulman et al., 2017) to multi-agent settings.

**Value Factorization Methods**  The second category of methods factorizes the joint value function into individual components. Monotonic factorization is an intuitive way to satisfy the IGM principle. For example, VDN (Sunehag et al., 2017) represents the joint action value as the sum of individual action values. QMIX (Rashid et al., 2020b) represents the joint action value with a monotonic network that uses a positive weighted linear to deal with individual action values. Qattan (Yang et al., 2020) uses multi-head attention (Vaswani et al., 2017) to generate mixing weights. NA$^2$Q (Liu et al., 2023) improves interpretability through Taylor expansion-based monotonic factorization. However, monotonic factorizations may suffer from representational limitations. To address this issue, QTRAN (Son et al., 2019) and WQMIX (Rashid et al., 2020a) pay more attention to the estimation of greedy action values. Specifically, QTRAN introduces a compensatory base to relax the non-optimal errors, while WQMIX reduces the weight on approximation of non-optimal parts. Other methods like QPLEX (Wang et al., 2021) and ResQ (Shen et al., 2022) design complete factorizations of the joint action value. QPLEX uses the maximum of a mixer value and a non-positive advantage function. ResQ combines a monotonic function with a non-positive function. GVR (Wan et al., 2022) takes a different approach by making $u^*$ the only stable point, although this requires numerous approximations.

**Communication-based Methods**  Other researchers study communication mechanisms in multi-agent systems. CommNet (Sukhbaatar et al., 2016) enables multi-round communication for sharing observations. DIAL (Foerster et al., 2016) designs gradient-based messages exchanged between agents. To decide with whom to communicate, TarMAC (Das et al., 2019) uses a signature-based soft attention mechanism, and MAGIC (Niu et al., 2021) uses graph neural networks. NDQ (Wang et al., 2020) combines value factorization and communication by mixing individual $Q$s through communication. ACE (Li et al., 2023) models communication as intermediate processes in the Markov Decision Process. In addition to parallel decision making, recent methods study sequential decision making through communication. PG-AR (Fu et al., 2022) randomizes the order of decision making. SeqComm (Ding et al., 2024) use intentions to prioritize agents in sequential decision-making.

**Other MARL Methods**  Recent works introduce additional innovations in MARL. RES (Pan et al., 2021) addresses value overestimation using regularized softmax losses. MAVEN (Mahajan et al., 2019) enhances exploration efficiency through randomized latent vectors. LIGS (Mguni et al., 2022) proposes learnable intrinsic rewards to improve multi-agent cooperation. SHAQ (Wang et al., 2022) integrates the Shapley value by modeling Q functions as marginal contributions of agents.

## A.2  RELATIONSHIP TO EXISTING WORK

**Relationship to Weighted QMIX**  Both Weighted QMIX (Rashid et al., 2020a) and the single-round value factorization in our method aim to enhance convergence toward actions superior to a given action. Weighted QMIX achieves this by reducing the fitting weight for inferior actions,

whereas our method assigns them the minimum target value. However, Weighted QMIX cannot guarantee strict improvement over the given action, as suboptimal actions may be stable points. In contrast, our method ensures strict improvement by preventing actions with minimum target value from becoming stable points. Moreover, even when Weighted QMIX obtains the optimal action, it may fail to stabilize at this optimum because the optimal action might not be a stable point. Although our method similarly risks deviating from the optimal action, we mitigate this by early termination when no further improvement is detected.

**Relationship to GVR**    The main idea of GVR (Wan et al., 2022) is to establish the optimal action as a stable point while rendering other actions non-stable. However, this approach requires a prior knowledge of the optimal action, which is fundamentally infeasible in MARL since finding the optimal action is the primary objective. Consequently, GVR inevitably relies on approximations of ideal conditions, which not only increase the complexity but may also convert certain suboptimal actions into stable points. In contrast to GVR, our method converts inferior actions into unstable points in each iteration. Determining inferior actions is computationally straightforward, and by assigning them the minimum target value, existing monotonic value factorizations can effectively exclude these inferior actions. Therefore, our method strictly improves the current action through iterations, providing a more reliable approximation to the optimal action. Furthermore, GVR only defines the concept of stable points within the policy distribution, and its analysis is limited to a particular form of QMIX. In contrast, we not only explicitly identify the origin of stable points (characterized by the lag between changes in non-differential variables and gradient-based parameters) but also provide a formal definition. In addition, we provide a comprehensive analysis on existing value factorization methods, unifying their suboptimal performance under a common explanation: the presence of at least one non-optimal stable point.

**Relationship to Communication**    Our method can also be interpreted as a multi-round communication algorithm within a parallel execution framework. In each iteration, agents exchange pre-decision information and generate new decisions through communication. The key challenge in such frameworks is ensuring that post-communication actions strictly improve upon pre-communication actions - failure to achieve this prevents agents from reaching consensus, resulting in suboptimal solutions. In our method, by rendering actions inferior to pre-communication ones as non-stable points, our approach not only guarantees the improvement of post-communication actions but also approaches the optimal action within sufficient communication rounds. In contrast, sequential execution frameworks essentially extend single-step action selection across agents, where each agent's action depends on higher-priority agents' choices. Such frameworks not only demonstrate lower efficiency but also introduce additional concerns regarding policy convergence and suboptimality caused by execution ordering (Ding et al., 2024).

# B    DEFINITION

In this section, we will define some important concepts that have been used in previous work but have not yet been rigorously defined.

## B.1    MONOTONIC PAYOFF

Note that the definition of monotonicity for payoffs differs from that in functions of real variables, as there is no ordering relationship defined in the action space, the domain of the payoff. We define an order relation on the actions based on the function $F$ as follows:

**Definition 2** (Order Relation $\succeq$). *Let $u_i^1, u_i^2 \in \mathcal{U}_i$ be two actions of agent $i$. For the function $F$, if the order relation between $u_i^1$ and $u_i^2$ is $u_i^1 \succeq u_i^2$, then*

$$F(u_i^1, \boldsymbol{u}_{-i}) \geq F(u_i^2, \boldsymbol{u}_{-i}), \forall \boldsymbol{u}_{-i} \in \boldsymbol{\mathcal{U}}_{-i} \tag{9}$$

*where $\boldsymbol{u}_{-i} = (u_1, \cdots, u_{i-1}, u_{i+1}, \cdots, u_n)$ is the joint action without $u_i$.*

The function $F$ can be replaced with $Q_{\mathrm{jt}}$, a measurement of payoff. However, the order relation based on $Q_{\mathrm{jt}}$ exists over the entire action space only if $Q_{\mathrm{jt}}$ is monotonic, which is defined as follows:

**Definition 3** (Monotonic $Q_{\mathrm{jt}}$). *$\forall \boldsymbol{s} \in \mathcal{S}$, if for every agent $i$ and any two actions $u_i^1, u_i^2 \in \mathcal{U}_i$, there exists an order relation that is either $u_i^1 \succeq u_i^2$ or $u_i^2 \succeq u_i^1$, then $Q_{\mathrm{jt}}$ is monotonic.*

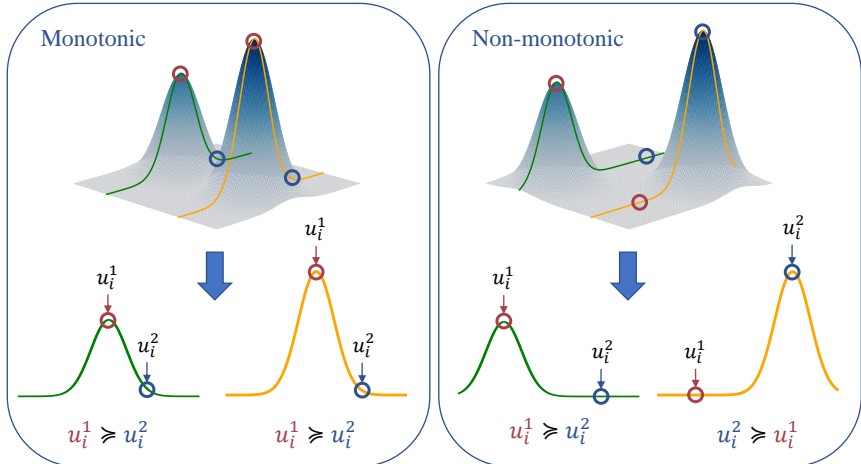

Figure 8: An illustration of Definition 3. We intercept curves corresponding to different $\boldsymbol{u}_{-i}$ and check whether their monotonicity with respect to $u_i$ is consistent.

Table 4 shows the payoff matrices with different monotonicity. Monotonic factorization methods can easily obtain the optimal solution in monotonic cases ((a) and (b) in Table 4) and sometimes in non-monotonic cases ((c) in Table 4). However, monotonic factorizations struggle to obtain the optimal solution in highly non-monotonic cases ((d) in Table 4).

| 9 | 8 | 7 |
|---|---|---|
| 6 | 5 | 4 |
| 3 | 2 | 1 |

(a) Monotonic

| 8 | 9 | 7 |
|---|---|---|
| 3 | 2 | 1 |
| 5 | 6 | 4 |

(b) Monotonic

| 9 | 0 | 0 |
|---|---|---|
| 0 | 5 | 0 |
| 0 | 0 | 0 |

(c) Non-monotonic

| 9 | -9 | -9 |
|---|---|---|
| -9 | 5 | 0 |
| -9 | 0 | 0 |

(d) Highly non-monotonic

Table 4: Payoff matrices with different monotonicity, where the value in row $r$ and column $c$ denotes the payoff of joint action $\boldsymbol{u} = (r, c)$. (a): A monotonic payoff matrix. (b) A monotonic payoff matrix generated by rearranging the rows and columns of (a). (c) A non-monotonic payoff matrix, where the relation between any two rows or columns is partially ordered. (d) A highly non-monotonic payoff matrix, where the relation between any two rows or columns is largely unordered.

## B.2 IDEAL QMIX

The objective of QMIX is to find individual $Q$s and $Q_{\text{mon}}$ that minimize the MSE between $Q_{\text{tot}}$ and $Q_{\text{jt}}$, which is defined in Equation (10) for certain $\boldsymbol{s} \in \mathcal{S}$ [3].

$$L_{\text{tot}}(\boldsymbol{s}) = \sum_{\boldsymbol{u}} \boldsymbol{\pi}(\boldsymbol{u}|\boldsymbol{s})(Q_{\text{mon}}(\boldsymbol{s}, \boldsymbol{u}) - Q_{\text{jt}}(\boldsymbol{s}, \boldsymbol{u}))^2 \tag{10}$$

Since the relationship between individual $Q$s and $Q_{\text{mon}}$ is monotonic, the magnitude relationship of $Q_{\text{mon}}$ is constrained by that of individual $Q$s, which is $\forall i \in \{1, 2, \cdots, n\}, \forall u_i, v_i \in \mathcal{U}_i, \boldsymbol{u}_{-i} \in \boldsymbol{\mathcal{U}}_{-i} = \mathcal{U}_1 \times \cdots \mathcal{U}_{i-1} \times \mathcal{U}_{i+1} \times \cdots \times \mathcal{U}_n$

$$(Q_{\text{mon}}(\boldsymbol{s}, \boldsymbol{u}_{-i}, u_i) - Q_{\text{mon}}(\boldsymbol{s}, \boldsymbol{u}_{-i}, v_i))(Q_i(\boldsymbol{s}, u_i) - Q_i(\boldsymbol{s}, v_i)) \geq 0 \tag{11}$$

Combining the objective defined in Equation (10) and the constrain defined in Equation (11), we get the ideal optimization problem that QMIX solves. In practice, QMIX uses the network as a monotonic function $f_{\text{mon}}$ to represent the constraint between individual $Q$s and $Q_{\text{mon}}$, and applies stochastic gradient descent (SGD) to minimize the objective. In this way, $Q_i$ is updated based on its gradient which involves $\frac{\partial f_{\text{mon}}}{\partial Q_i}$. However, since how $\frac{\partial f_{\text{mon}}}{\partial Q_i}$ varies is unknown, to avoid this confounding factor, we assume that QMIX can obtain the optimal combination of individual $Q$s and

---

[3]To simplify the discussion, we focus on the fully observable environment by replacing $\boldsymbol{\tau}$ with $\boldsymbol{s}$ in $Q_{\text{tot}}$

$Q_{\mathrm{mon}}$ that satisfies Equation (11), and consequently minimize $L_{\mathrm{tot}}$. We name QMIX based on this assumption as the ideal QMIX which is defined as follows [4]:

**Definition 4** (Ideal QMIX). *Let $\mathcal{Q}$ be the set of pairs $(\boldsymbol{Q}, Q_{\mathrm{mon}})$, where the individual action values $\boldsymbol{Q} = (Q_1, Q_2, \cdots, Q_n)$ and $Q_{\mathrm{mon}}$ satisfy Equation (11). For a joint action value $Q_{\mathrm{jt}}$ and policy $\boldsymbol{\pi}$, $\boldsymbol{Q}$ and $Q_{\mathrm{mon}}$ of the ideal QMIX converge to $\boldsymbol{Q}^*$ and $Q_{\mathrm{mon}}^*$ which satisfy $\forall \boldsymbol{s} \in \mathcal{S}$*

$$\boldsymbol{Q}^*, Q_{\mathrm{mon}}^* \in \operatorname*{arg\,min}_{(\boldsymbol{Q}, Q_{\mathrm{mon}}) \in \mathcal{Q}} \sum_{\boldsymbol{u}} \boldsymbol{\pi}(\boldsymbol{u}|\boldsymbol{s})(Q_{\mathrm{mon}}(\boldsymbol{s}, \boldsymbol{u}) - Q_{\mathrm{jt}}(\boldsymbol{s}, \boldsymbol{u}))^2 \tag{12}$$

*where $Q_{\mathrm{tot}} = Q_{\mathrm{mon}}$ in QMIX.*

As for other value factorization methods such as WQMIX (Rashid et al., 2020a) and ResQ (Shen et al., 2022) where $Q_{\mathrm{mon}}$ appears in their design, we also assume they share the same property as the ideal QMIX, which aligns with the assumption used in their papers.

From Definition 4, the ideal QMIX obtains a monotonic representation of $Q_{\mathrm{jt}}$ with the minimum $L_{\mathrm{tot}}$. We illustrate this process in matrix cases: Given the order of rows and columns, we can find the monotonic matrix with local minimum $L_{\mathrm{tot}}$ by solving the constraint optimization problem. To find the monotonic matrix with minimum $L_{\mathrm{tot}}$, we compare $L_{\mathrm{tot}}$ for all possible orders. We give an example of the ideal QMIX shown in Table 5.

| 8 | -12 | -12 |
|---|---|---|
| -12 | 3 | 0 |
| -12 | 0 | 5 |

(a) $Q_{\mathrm{jt}}$

$L_{\mathrm{tot}}=36.22$

| -8 | -8 | -8 |
|---|---|---|
| -8 | 1 | 1 |
| -8 | 1 | **5** |

(b) $Q_{\mathrm{tot}}^*$

$L_{\mathrm{tot}}=45.56$

| **8** | -5 | -5 |
|---|---|---|
| -5 | -5 | -5 |
| -5 | -5 | -5 |

(c) $Q_{\mathrm{tot}}^*$ for $\overline{\boldsymbol{u}} = \boldsymbol{u}^*$

Table 5: (a): A non-monotonic payoff matrix, where the value presented in row $r$ and column $c$ denotes the $Q_{\mathrm{jt}}$ value of joint action $(r, c)$. (b): The monotonic representation with the global minimum $L_{\mathrm{tot}}$. (c) The optimal monotonic representation constrained by $\overline{\boldsymbol{u}} = \boldsymbol{u}^*$. Since the $L_{\mathrm{tot}}$ of (b) is less than the $L_{\mathrm{tot}}$ of (c), $Q_{\mathrm{tot}}$ of the ideal QMIX under uniform visitation ($\epsilon = 1$) will converge to (b), which fails to get the global optimal action.

## C  PROOF

### C.1  THEOREM ON MRVF

**Theorem 1.** *For a joint action value $Q_{\mathrm{jt}}$ with a finite action space $\mathcal{U}$ and $\boldsymbol{s} \in \mathcal{S}$, consider a sequence $\{\overline{\boldsymbol{u}}^k | \overline{\boldsymbol{u}}^k \in \mathcal{U}\}$, if $\forall k > 1, Q_{\mathrm{jt}}(\boldsymbol{s}, \overline{\boldsymbol{u}}^k) > Q_{\mathrm{jt}}(\boldsymbol{s}, \overline{\boldsymbol{u}}^{k-1})$ when $\overline{\boldsymbol{u}}^{k-1} \neq \boldsymbol{u}^*$, then $\exists K > 0, \overline{\boldsymbol{u}}^K = \boldsymbol{u}^*$.*

*Proof.* Let $\mathcal{U}_+^k = \{\boldsymbol{u} | Q_{\mathrm{jt}}(\boldsymbol{s}, \boldsymbol{u}) > Q_{\mathrm{jt}}(\boldsymbol{s}, \overline{\boldsymbol{u}}^k), \boldsymbol{u} \in \mathcal{U}\}$. **Assume** that $\forall k > 0, \overline{\boldsymbol{u}}^k \neq \boldsymbol{u}^*$. Since $\forall k > 1, Q_{\mathrm{jt}}(\boldsymbol{s}, \overline{\boldsymbol{u}}^k) > Q_{\mathrm{jt}}(\boldsymbol{s}, \overline{\boldsymbol{u}}^{k-1})$ when $\overline{\boldsymbol{u}}^{k-1} \neq \boldsymbol{u}^*, \mathcal{U}_+^k \subset \mathcal{U}_+^{k-1}$. Therefore, we have $|\mathcal{U}_+^{k-1}| - |\mathcal{U}_+^k| \geq 1$. Then, we have $\forall k > 0, |\mathcal{U}_+^k| \leq |\mathcal{U}_+^1| - (k - 1)$. If we choose $K > |\mathcal{U}| + 1$, since $\mathcal{U}_+^1 \subseteq \mathcal{U}$, we get $|\mathcal{U}_+^K| < |\mathcal{U}_+^1| - |\mathcal{U}| \leq 0$. Since $\forall k > 0, |\mathcal{U}_+^k| \geq 0$, the assumption is false. Therefore, we prove that $\exists K > 0, \overline{\boldsymbol{u}}^K = \boldsymbol{u}^*$. □

**Lemma 1.** *For the converged $Q_{\mathrm{tot}}$ of the idea QMIX defined in Equation (12) and $\epsilon$-greedy policy $\boldsymbol{\pi}$ that is continuous for $\epsilon \in [0, 1]$ and satisfies Equation (13), we have $\lim_{\epsilon \to 0^+} Q_{\mathrm{tot}}^*(\boldsymbol{s}, \widetilde{\boldsymbol{u}}) = Q_{\mathrm{jt}}(\boldsymbol{s}, \widetilde{\boldsymbol{u}})$ for $\boldsymbol{s} \in \mathcal{S}$.*

$$\lim_{\epsilon \to 0^+} \boldsymbol{\pi}(\boldsymbol{u}|\boldsymbol{s}) = \begin{cases} 1 & \boldsymbol{u} = \widetilde{\boldsymbol{u}} \\ 0 & otherwise \end{cases} \tag{13}$$

---

[4]The policy $\boldsymbol{\pi}$ is omitted in the definition by Rashid et al. (2020a). However, it plays a crucial role in shaping $Q_{\mathrm{tot}}$ and will be extensively analyzed in our proof.

*Proof.* **Assume** that $\exists \Delta > 0, \forall E > 0, \exists \epsilon \in (0, E), |Q_{\text{tot}}^*(\boldsymbol{s}, \widetilde{\boldsymbol{u}}) - Q_{\text{jt}}(\boldsymbol{s}, \widetilde{\boldsymbol{u}})| \geq \Delta$ The loss of $Q_{\text{tot}}^*$ defined in Equation (12) is

$$L_{\text{tot}}^* = \sum_{\boldsymbol{u}} \boldsymbol{\pi}(\boldsymbol{u}|\boldsymbol{s})(Q_{\text{tot}}^*(\boldsymbol{s}, \boldsymbol{u}) - Q_{\text{jt}}(\boldsymbol{s}, \boldsymbol{u}))^2$$

$$= \sum_{\boldsymbol{u} \neq \widetilde{\boldsymbol{u}}} \boldsymbol{\pi}(\boldsymbol{u}|\boldsymbol{s})(Q_{\text{tot}}^*(\boldsymbol{s}, \boldsymbol{u}) - Q_{\text{jt}}(\boldsymbol{s}, \boldsymbol{u}))^2 + \boldsymbol{\pi}(\widetilde{\boldsymbol{u}}|\boldsymbol{s})(Q_{\text{tot}}^*(\boldsymbol{s}, \widetilde{\boldsymbol{u}}) - Q_{\text{jt}}(\boldsymbol{s}, \widetilde{\boldsymbol{u}}))^2 \quad (14)$$

$$\geq \boldsymbol{\pi}(\widetilde{\boldsymbol{u}}|\boldsymbol{s})\Delta^2$$

Let $Q_{\text{tot}}'(\boldsymbol{s}, \cdot) \equiv \frac{\Delta}{2} + Q_{\text{jt}}(\boldsymbol{s}, \widetilde{\boldsymbol{u}})$. The loss of $Q_{\text{tot}}'$ is

$$L_{\text{tot}}' = \sum_{\boldsymbol{u} \neq \widetilde{\boldsymbol{u}}} \boldsymbol{\pi}(\boldsymbol{u}|\boldsymbol{s})(\frac{\Delta}{2} + Q_{\text{jt}}(\boldsymbol{s}, \widetilde{\boldsymbol{u}}) - Q_{\text{jt}}(\boldsymbol{s}, \boldsymbol{u}))^2 + \boldsymbol{\pi}(\widetilde{\boldsymbol{u}}|\boldsymbol{s})(\frac{\Delta}{2})^2 \quad (15)$$

Consider $L_{\text{tot}}^* - L_{\text{tot}}'$, we have

$$L_{\text{tot}}^* - L_{\text{tot}}' \geq \boldsymbol{\pi}(\widetilde{\boldsymbol{u}}|\boldsymbol{s})(\frac{\Delta}{2})^2 - \sum_{\boldsymbol{u} \neq \widetilde{\boldsymbol{u}}} \boldsymbol{\pi}(\boldsymbol{u}|\boldsymbol{s})(\frac{\Delta}{2} + Q_{\text{jt}}(\boldsymbol{s}, \widetilde{\boldsymbol{u}}) - Q_{\text{jt}}(\boldsymbol{s}, \boldsymbol{u}))^2 \quad (16)$$

Notice that

$$\lim_{\epsilon \to 0^+} (\boldsymbol{\pi}(\widetilde{\boldsymbol{u}}|\boldsymbol{s})(\frac{\Delta}{2})^2 - \sum_{\boldsymbol{u} \neq \widetilde{\boldsymbol{u}}} \boldsymbol{\pi}(\boldsymbol{u}|\boldsymbol{s})(\frac{\Delta}{2} + Q_{\text{jt}}(\boldsymbol{s}, \widetilde{\boldsymbol{u}}) - Q_{\text{jt}}(\boldsymbol{s}, \boldsymbol{u}))^2) = (\frac{\Delta}{2})^2 > 0 \quad (17)$$

Therefore, $\exists E > 0, \forall \epsilon \in (0, E), L_{\text{tot}}^* > L_{\text{tot}}'$ which is against with condition that $Q_{\text{tot}}^*$ satisfies Equation (12). Thus, the assumption is false. We have $\forall \Delta > 0, \exists E > 0, \forall \epsilon \in (0, E), |Q_{\text{tot}}^*(\boldsymbol{s}, \widetilde{\boldsymbol{u}}) - Q_{\text{jt}}(\boldsymbol{s}, \widetilde{\boldsymbol{u}})| < \Delta$ which means $\lim_{\epsilon \to 0^+} Q_{\text{tot}}^*(\boldsymbol{s}, \widetilde{\boldsymbol{u}}) = Q_{\text{jt}}(\boldsymbol{s}, \widetilde{\boldsymbol{u}})$. $\square$

**Theorem 2.** *Consider a non-constant joint action value $Q_{\text{jt}}(\boldsymbol{s}, \cdot)$ ($\forall \boldsymbol{s} \in \mathcal{S}, \forall C \in \mathbb{R}, Q_{\text{jt}}(\boldsymbol{s}, \cdot) \not\equiv C$) with finite state space $\mathcal{S}$ and action space $\mathcal{U}$. For the ideal QMIX defined in Equation (12) and the centralized $\epsilon$-greedy policy $\boldsymbol{\pi}$ defined in Equation (8), we have $\forall \boldsymbol{s} \in \mathcal{S}, \exists E \in (0, 1), \forall \epsilon \in (0, E), \forall \boldsymbol{u} \in \arg\min_{\boldsymbol{u}} Q_{\text{jt}}(\boldsymbol{s}, \boldsymbol{u}), \boldsymbol{u}$ is not a stable point.*

*Proof.* For $\boldsymbol{s} \in \mathcal{S}$, **Assume** that $\forall E_s \in (0, 1], \exists \epsilon \in (0, E_s), \exists Q_{\text{tot}}^*$ that $Q_{\text{jt}}(\boldsymbol{s}, \overline{\boldsymbol{u}}^*) = \min_{\boldsymbol{u}} Q_{\text{jt}}(\boldsymbol{s}, \boldsymbol{u})$ and $\widetilde{\boldsymbol{u}} = \overline{\boldsymbol{u}}^*$ ($\overline{\boldsymbol{u}}^*$ is a stable point). We define $\mathcal{U}^+ = \{\boldsymbol{u}|Q_{\text{jt}}(\boldsymbol{s}, \boldsymbol{u}) > \min_{\boldsymbol{u}} Q_{\text{jt}}(\boldsymbol{s}, \boldsymbol{u})\}$ and $\mathcal{U}^- = \mathcal{U} - \mathcal{U}^+$ . Since $\lim_{\epsilon \to 0^+} Q_{\text{tot}}^*(\boldsymbol{s}, \overline{\boldsymbol{u}}^*) = Q_{\text{jt}}(\boldsymbol{s}, \overline{\boldsymbol{u}}^*)$ (Lemma 1 and $\widetilde{\boldsymbol{u}} = \overline{\boldsymbol{u}}^*$), we have $\exists E_s^0 \in (0, 1], \forall \epsilon \in (0, E_s^0), Q_{\text{tot}}^*(\boldsymbol{s}, \overline{\boldsymbol{u}}^*) < \min_{\boldsymbol{u} \in \mathcal{U}^+} Q_{\text{jt}}(\boldsymbol{s}, \boldsymbol{u})$. Thus, $\forall \boldsymbol{u} \in \mathcal{U}^+$, since $Q_{\text{tot}}^*(\boldsymbol{s}, \boldsymbol{u}) \leq Q_{\text{tot}}^*(\boldsymbol{s}, \overline{\boldsymbol{u}}^*) < Q_{\text{jt}}(\boldsymbol{s}, \boldsymbol{u})$, consider a term of $L_{\text{tot}}^*$, we have

$$L_{\text{tot}}^*(\boldsymbol{u}) = \boldsymbol{\pi}(\boldsymbol{u}|\boldsymbol{s})(Q_{\text{tot}}^*(\boldsymbol{s}, \boldsymbol{u}) - Q_{\text{jt}}(\boldsymbol{s}, \boldsymbol{u}))^2$$
$$> \frac{\epsilon}{|\mathcal{U}|}(Q_{\text{tot}}^*(\boldsymbol{s}, \overline{\boldsymbol{u}}^*) - Q_{\text{jt}}(\boldsymbol{s}, \boldsymbol{u})) \quad (18)$$

Therefore, for the loss of $Q_{\text{tot}}^*$ defined in Equation (12), we have

$$L_{\text{tot}}^* = \sum_{\boldsymbol{u}} \boldsymbol{\pi}(\boldsymbol{u}|\boldsymbol{s})(Q_{\text{tot}}^*(\boldsymbol{s}, \boldsymbol{u}) - Q_{\text{jt}}(\boldsymbol{s}, \boldsymbol{u}))^2$$
$$> \frac{\epsilon}{|\mathcal{U}|} \sum_{\boldsymbol{u} \in \mathcal{U}^+} (Q_{\text{tot}}^*(\boldsymbol{s}, \overline{\boldsymbol{u}}^*) - Q_{\text{jt}}(\boldsymbol{s}, \boldsymbol{u}))^2 + (1 - \epsilon + \frac{\epsilon}{|\mathcal{U}|})(Q_{\text{tot}}^*(\boldsymbol{s}, \overline{\boldsymbol{u}}^*) - Q_{\text{jt}}(\boldsymbol{s}, \overline{\boldsymbol{u}}^*))^2 \quad (19)$$

Consider certain $\boldsymbol{u}^+ \in \mathcal{U}^+$. We define $Q_{\text{tot}}'$ as

$$Q_{\text{tot}}'(\boldsymbol{s}, \boldsymbol{u}) = \begin{cases} Q_{\text{jt}}(\boldsymbol{s}, \boldsymbol{u}^+) & \boldsymbol{u} = \boldsymbol{u}^+ \\ Q_{\text{tot}}^*(\boldsymbol{s}, \overline{\boldsymbol{u}}^*) & otherwise \end{cases} \quad (20)$$

The loss of $Q_{\text{tot}}'$ defined in Equation (12) is ($\widetilde{\boldsymbol{u}} = \overline{\boldsymbol{u}}^*$)

$$L_{\text{tot}}' = \frac{\epsilon}{|\mathcal{U}|}(\sum_{\boldsymbol{u} \in (\mathcal{U}^+ - \{\boldsymbol{u}^+\})} (Q_{\text{tot}}^*(\boldsymbol{s}, \overline{\boldsymbol{u}}^*) - Q_{\text{jt}}(\boldsymbol{s}, \boldsymbol{u}))^2 + \sum_{\boldsymbol{u} \in \mathcal{U}^-} (Q_{\text{tot}}^*(\boldsymbol{s}, \overline{\boldsymbol{u}}^*) - Q_{\text{jt}}(\boldsymbol{s}, \overline{\boldsymbol{u}}^*))^2)$$

$$+ (1 - \epsilon + \frac{\epsilon}{|\mathcal{U}|})(Q_{\text{tot}}^*(\boldsymbol{s}, \overline{\boldsymbol{u}}^*) - Q_{\text{jt}}(\boldsymbol{s}, \overline{\boldsymbol{u}}^*))^2$$

$$(21)$$

Comparing $L_{\text{tot}}^*$ and $L_{\text{tot}}^{'}$, we have

$$L_{\text{tot}}^* - L_{\text{tot}}^{'} > \frac{\epsilon}{|\mathcal{U}|}((Q_{\text{tot}}^*(\boldsymbol{s}, \overline{\boldsymbol{u}}^*) - Q_{\text{jt}}(\boldsymbol{s}, \boldsymbol{u}^+))^2 - |\mathcal{U}^-|(Q_{\text{tot}}^*(\boldsymbol{s}, \overline{\boldsymbol{u}}^*) - Q_{\text{jt}}(\boldsymbol{s}, \overline{\boldsymbol{u}}^*))^2) \quad (22)$$

Notice that

$$f(x) = (x - Q_{\text{jt}}(\boldsymbol{s}, \boldsymbol{u}^+))^2 - |\mathcal{U}^-|(x - Q_{\text{jt}}(\boldsymbol{s}, \overline{\boldsymbol{u}}^*))^2 \quad (23)$$

is a continuous function and $\lim_{x \to Q_{\text{jt}}(\boldsymbol{s}, \overline{\boldsymbol{u}}^*)} f(x) = (Q_{\text{jt}}(\boldsymbol{s}, \overline{\boldsymbol{u}}^*) - Q_{\text{jt}}(\boldsymbol{s}, \boldsymbol{u}^+))^2 > 0$. Since $\lim_{\epsilon \to 0^+} Q_{\text{tot}}^*(\boldsymbol{s}, \overline{\boldsymbol{u}}^*) = Q_{\text{jt}}(\boldsymbol{s}, \overline{\boldsymbol{u}}^*)$, we have $\exists E_s^1 \in (0, 1], \forall \epsilon \in (0, E_s^1), f(Q_{\text{tot}}^*(\boldsymbol{s}, \overline{\boldsymbol{u}}^*)) > 0$. Therefore, we find $E_s = \min\{E_s^0, E_s^1\} \in (0, 1]$ that $\forall \epsilon \in (0, E_s)$ $L_{\text{tot}}^* > L_{\text{tot}}^{'}$, which is against with the condition that $Q_{\text{tot}}^*$ satisfies Equation (12). Thus, $\exists E_s \in (0, 1], \forall \epsilon \in (0, E_s), \forall \boldsymbol{u} \in \arg\min_{\boldsymbol{u}} Q_{\text{jt}}(\boldsymbol{s}, \boldsymbol{u})$, $\boldsymbol{u}$ is not a stable point. Finally, we complete the proof by taking $E = \min_{s \in \mathcal{S}} E_s$. $\square$

## C.2 THEOREM ON RESQ

We will prove that $\boldsymbol{u}^*$ is the unique yet weakly stable point of ResQ (Shen et al., 2022). Here, we present the expression of $Q_{\text{tot}}$ of ResQ.

$$Q_{\text{tot}}(\boldsymbol{s}, \boldsymbol{u}) = Q_{\text{mon}}(\boldsymbol{s}, \boldsymbol{u}) + w_{\text{r}}(\boldsymbol{s}, \boldsymbol{u}) * Q_{\text{r}}(\boldsymbol{s}, \boldsymbol{u}) \quad (24)$$

where $Q_{\text{mon}} = f_{\text{mon}}(Q_1, Q_2, \cdots, Q_n), Q_{\text{r}} \leq 0$ and

$$w_{\text{r}}(\boldsymbol{s}, \boldsymbol{u}) = \begin{cases} 0 & \boldsymbol{u} = \widetilde{\boldsymbol{u}} \\ 1 & otherwise \end{cases} \quad (25)$$

We provide a specific instance for the weak stability of ResQ in Table 6. Then, we prove that the weak stability occurs in general cases in Theorem 3.

| 8 | -12 | -12 |
|---|---|---|
| -12 | 3 | 0 |
| -12 | 0 | 5 |

(a) $Q_{\text{jt}}$

| 8 | 8 | 8 |
|---|---|---|
| 8 | 8 | 9 |
| 8 | 8 | 8 |

(b) $Q_{\text{mon}}$

| 0 | -20 | -20 |
|---|---|---|
| -20 | -5 | -9 |
| -20 | -8 | -4 |

(c) $w_{\text{r}} * Q_{\text{r}}$

Table 6: This table shows the weak stability of ResQ for a specific $Q_{\text{jt}}$ in (a), where we let $\widetilde{\boldsymbol{u}} = \boldsymbol{u}^*$ initially to check whether ResQ will stabilize at $\boldsymbol{u}^*$. We highlight the values related to $\widetilde{\boldsymbol{u}}$ in red and $\overline{\boldsymbol{u}}$ in blue. $Q_{\text{tot}}$ is the sum of $Q_{\text{mon}}$ in (b) and $w_{\text{r}} * Q_{\text{r}}$ in (c), which is identical to $Q_{\text{jt}}$ ($L_{\text{tot}} = 0$). However, since $\overline{\boldsymbol{u}} \neq \boldsymbol{u}^*$ may happens, $\boldsymbol{u}^*$ is not a strongly stable point for ResQ.

**Lemma 2.** *For a joint action value $Q_{\text{jt}}$ and $\boldsymbol{s} \in \mathcal{S}, \forall \widetilde{\boldsymbol{u}} \in \mathcal{U}$, we can find $Q_{\text{tot}}$ with $L_{\text{tot}} = 0$ and $\overline{\boldsymbol{u}} \neq \widetilde{\boldsymbol{u}}$.*

*Proof.* We can find a $Q_{\text{mon}}$ that satisfies

(1) $Q_{\text{mon}} \geq Q_{\text{jt}}$

(2) $Q_{\text{mon}}(\boldsymbol{s}, \widetilde{\boldsymbol{u}}) = Q_{\text{jt}}(\boldsymbol{s}, \widetilde{\boldsymbol{u}})$

(3) $\overline{\boldsymbol{u}} \neq \widetilde{\boldsymbol{u}}$

Here we give a specific $Q_{\text{mon}}$ that satisfies these conditions:

$$Q_{\text{mon}}(\boldsymbol{s}, \boldsymbol{u}) = \begin{cases} Q_{\text{jt}}(\boldsymbol{s}, \widetilde{\boldsymbol{u}}) & \boldsymbol{u} = \widetilde{\boldsymbol{u}} \\ \max_{\boldsymbol{u}} Q_{\text{jt}}(\boldsymbol{s}, \boldsymbol{u}) + \Delta & otherwise \end{cases} \quad (26)$$

where $\Delta > 0$. We let $Q_{\text{r}} = Q_{\text{jt}} - Q_{\text{mon}}$. Since $Q_{\text{mon}} \geq Q_{\text{jt}}$, we have $Q_{\text{r}} \leq 0$.

$\forall \boldsymbol{u} \neq \widetilde{\boldsymbol{u}}$, we have $Q_{\text{jt}}(\boldsymbol{s}, \boldsymbol{u}) = Q_{\text{tot}}(\boldsymbol{s}, \boldsymbol{u}) = Q_{\text{mon}}(\boldsymbol{s}, \boldsymbol{u}) + 1 * Q_{\text{r}}(\boldsymbol{s}, \boldsymbol{u})$. For $\widetilde{\boldsymbol{u}}$, since $Q_{\text{mon}}(\boldsymbol{s}, \widetilde{\boldsymbol{u}}) = Q_{\text{jt}}(\boldsymbol{s}, \widetilde{\boldsymbol{u}})$, we have $Q_{\text{jt}}(\boldsymbol{s}, \widetilde{\boldsymbol{u}}) = Q_{\text{tot}}(\boldsymbol{s}, \widetilde{\boldsymbol{u}}) = Q_{\text{mon}}(\boldsymbol{s}, \widetilde{\boldsymbol{u}}) + 0 * Q_{\text{r}}(\boldsymbol{s}, \widetilde{\boldsymbol{u}})$.

Therefore, $\forall \boldsymbol{u} \in \mathcal{U}$, we have $Q_{\text{jt}}(\boldsymbol{s}, \boldsymbol{u}) = Q_{\text{tot}}(\boldsymbol{s}, \boldsymbol{u})$, which means $L_{\text{tot}} = 0$. Thus, we find $Q_{\text{tot}}$ with $L_{\text{tot}} = 0$ and $\overline{\boldsymbol{u}} \neq \widetilde{\boldsymbol{u}}$. $\square$

**Theorem 3.** *For a joint action value $Q_{\mathrm{jt}}$ and $s \in \mathcal{S}$, $\forall \boldsymbol{u}^* \in \arg\max_{\boldsymbol{u}} Q_{\mathrm{jt}}(s, \boldsymbol{u})$, $\boldsymbol{u}^*$ is a weakly stable point of ResQ (Shen et al., 2022), while $\forall \boldsymbol{u}^- \notin \arg\max_{\boldsymbol{u}} Q_{\mathrm{jt}}(s, \boldsymbol{u})$ is not a stable point of ResQ.*

*Proof.* First, we illustrate that $\boldsymbol{u}^*$ is a stable point. **Assume** that $\widetilde{\boldsymbol{u}} = \boldsymbol{u}^*$. We can find a $Q_{\mathrm{mon}}$ that satisfies

(1) $Q_{\mathrm{mon}} \geq Q_{\mathrm{jt}}$

(2) $Q_{\mathrm{mon}}(s, \widetilde{\boldsymbol{u}}) = Q_{\mathrm{jt}}(s, \widetilde{\boldsymbol{u}})$

(3) $\overline{\boldsymbol{u}} = \boldsymbol{u}^*$

where $Q_{\mathrm{mon}} \equiv Q_{\mathrm{jt}}(s, \boldsymbol{u}^*)$ is a specific case.

We let $Q_{\mathrm{r}} = Q_{\mathrm{jt}} - Q_{\mathrm{mon}}$. Since $Q_{\mathrm{mon}} \geq Q_{\mathrm{jt}}$, we have $Q_{\mathrm{r}} \leq 0$.

$\forall \boldsymbol{u} \neq \widetilde{\boldsymbol{u}}$, we have $Q_{\mathrm{jt}}(s, \boldsymbol{u}) = Q_{\mathrm{tot}}(s, \boldsymbol{u}) = Q_{\mathrm{mon}}(s, \boldsymbol{u}) + 1 * Q_{\mathrm{r}}(s, \boldsymbol{u})$. For $\widetilde{\boldsymbol{u}}$, since $Q_{\mathrm{mon}}(s, \widetilde{\boldsymbol{u}}) = Q_{\mathrm{jt}}(s, \widetilde{\boldsymbol{u}})$, we have $Q_{\mathrm{jt}}(s, \widetilde{\boldsymbol{u}}) = Q_{\mathrm{tot}}(s, \widetilde{\boldsymbol{u}}) = Q_{\mathrm{mon}}(s, \widetilde{\boldsymbol{u}}) + 0 * Q_{\mathrm{r}}(s, \widetilde{\boldsymbol{u}})$.

Therefore, we have $L_{\mathrm{tot}} = 0$, which is the lower bound of loss. However, according to Lemma 2, we can find $Q'_{\mathrm{tot}}$ with $L'_{\mathrm{tot}} = L_{\mathrm{tot}}$ while $\overline{\boldsymbol{u}}' \neq \boldsymbol{u}^*$. Thus, $\boldsymbol{u}^*$ is a weakly stable point of ResQ.

Second, we illustrate that $\boldsymbol{u}^-$ is not a stable point. Let $\widetilde{\boldsymbol{u}} = \boldsymbol{u}^-$, **Assume** that there exists $Q_{\mathrm{tot}}$ satisfying $\overline{\boldsymbol{u}} = \boldsymbol{u}^-$ and achieving the minimum loss. According to Lemma 2, we have $L_{\mathrm{tot}} = 0$, otherwise $L_{\mathrm{tot}}$ is not the minimum.

Since $L_{\mathrm{tot}} = 0$, we have $\forall \boldsymbol{u} \in \mathcal{U}, Q_{\mathrm{tot}}(s, \boldsymbol{u}) = Q_{\mathrm{jt}}(s, \boldsymbol{u})$. For $\boldsymbol{u}^-$, we have $Q_{\mathrm{tot}}(s, \boldsymbol{u}^-) = Q_{\mathrm{jt}}(s, \boldsymbol{u}^-)$. For $\boldsymbol{u}^*$, we have $Q_{\mathrm{tot}}(s, \boldsymbol{u}^*) = Q_{\mathrm{jt}}(s, \boldsymbol{u}^*)$.

However, since $Q_{\mathrm{tot}}(s, \boldsymbol{u}^-) \geq Q_{\mathrm{tot}}(s, \boldsymbol{u}^*)$ due to $\overline{\boldsymbol{u}} = \boldsymbol{u}^-$, we get a contradiction where $Q_{\mathrm{jt}}(s, \boldsymbol{u}^-) \geq Q_{\mathrm{jt}}(s, \boldsymbol{u}^*)$. Therefore, $\boldsymbol{u}^-$ is not a stable point.

$\square$

## D  STABILITY ANALYSIS OF QPLEX

In this section, we provide a suboptimal case of QPLEX in matrix games. Here, we present the expression of $Q_{\mathrm{tot}}$ of QPLEX.

$$Q_{\mathrm{tot}}(\boldsymbol{\tau}, \boldsymbol{u}) = \sum_i \max_{u_i} Q_i(\tau_i, u_i) + \sum_i w_i(\boldsymbol{\tau}, \boldsymbol{u}) * (Q_i(\tau_i, u_i) - \max_{u_i} Q_i(\tau_i, u_i)) \tag{27}$$

where $w_i(\boldsymbol{\tau}, \boldsymbol{u}) \geq 0$.

For QPLEX, stable point occurs when the gradients of $\boldsymbol{Q}$ and $\boldsymbol{w}$ satisfy that $\forall i \in \{1, 2, \cdots, N\}$

$$\frac{\partial L_{\mathrm{tot}}}{\partial \boldsymbol{Q}_i} = \boldsymbol{0}, \frac{\partial L_{\mathrm{tot}}}{\partial \boldsymbol{w}_i} \geq \boldsymbol{0} \tag{28}$$

where $\frac{\partial L_{\mathrm{tot}}}{\partial w_i(\boldsymbol{\tau}, \boldsymbol{u})} > 0$ only when $w_i(\boldsymbol{\tau}, \boldsymbol{u}) = 0$.

The suboptimal cases of QPLEX are presented in Table 7 and Table 8, where there are three stable points but only one is optimal. For readers to verify those stable points, we expand Equation (28) into Equation (29) and substitute $\widetilde{\boldsymbol{u}}$ with $\overline{\boldsymbol{u}}$.

$$\frac{\partial L_{\mathrm{tot}}}{\partial Q_i(\tau_i, u_i)} = \begin{cases} \sum\limits_{\boldsymbol{u} \in \mathcal{U}(\boldsymbol{u}_i = u_i)} (Q_{\mathrm{tot}}(\boldsymbol{\tau}, \boldsymbol{u}) - Q_{\mathrm{jt}}(\boldsymbol{\tau}, \boldsymbol{u})) w_i(\boldsymbol{\tau}, \boldsymbol{u}) & u_i \neq \overline{\boldsymbol{u}}_i \\[2em] \sum\limits_{\boldsymbol{u} \in \mathcal{U}(\boldsymbol{u}_i = u_i)} (Q_{\mathrm{tot}}(\boldsymbol{\tau}, \boldsymbol{u}) - Q_{\mathrm{jt}}(\boldsymbol{\tau}, \boldsymbol{u})) w_i(\boldsymbol{\tau}, \boldsymbol{u}) + \\ \sum\limits_{\boldsymbol{u} \in \mathcal{U}} (Q_{\mathrm{tot}}(\boldsymbol{\tau}, \boldsymbol{u}) - Q_{\mathrm{jt}}(\boldsymbol{\tau}, \boldsymbol{u}))(1 - w_i(\boldsymbol{\tau}, \boldsymbol{u})) & u_i = \overline{\boldsymbol{u}}_i \end{cases} \tag{29}$$

$$\frac{\partial L_{\mathrm{tot}}}{\partial w_i(\boldsymbol{\tau}, \boldsymbol{u})} = (Q_{\mathrm{tot}}(\boldsymbol{\tau}, \boldsymbol{u}) - Q_{\mathrm{jt}}(\boldsymbol{\tau}, \boldsymbol{u}))(Q_i(\tau_i, \boldsymbol{u}_i) - Q_i(\tau_i, \overline{\boldsymbol{u}}_i))$$

where $\mathcal{U}(\boldsymbol{u}_i = u_i) = \{\boldsymbol{u} | \boldsymbol{u} \in \mathcal{U}, \boldsymbol{u}_i = u_i\}$.

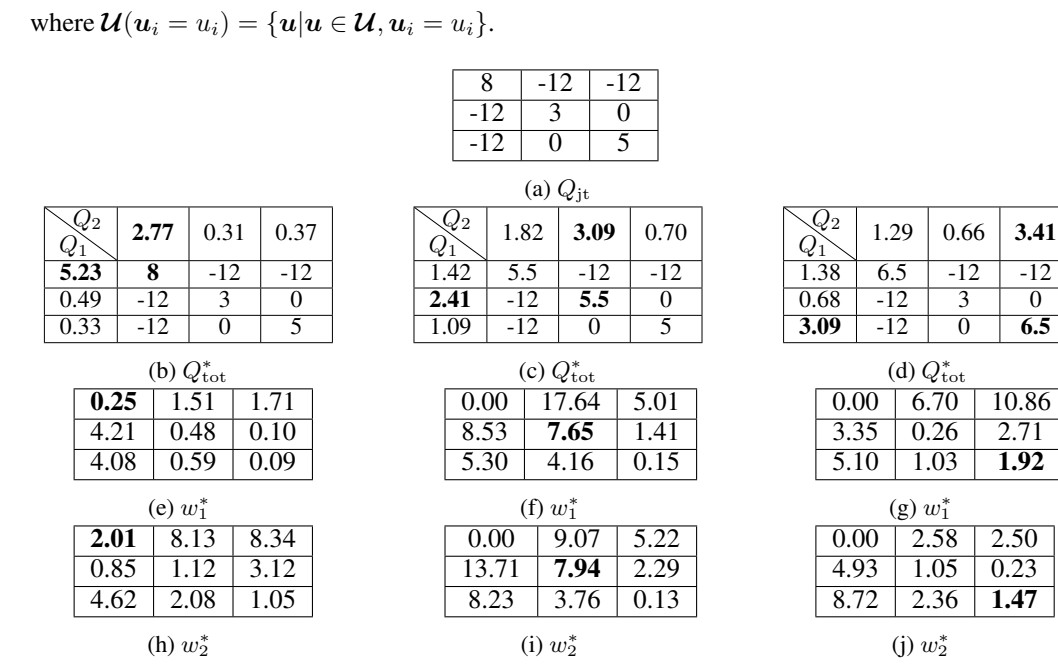

Table 7: The example of QPLEX in which $\boldsymbol{u}^*$ is not the only one stable points. The quantities of $Q_{\text{tot}}^*$ and $w_i^*$ listed in the same column correspond to a particular stable point, where we highlight $\overline{\boldsymbol{u}}^*$ in bold.

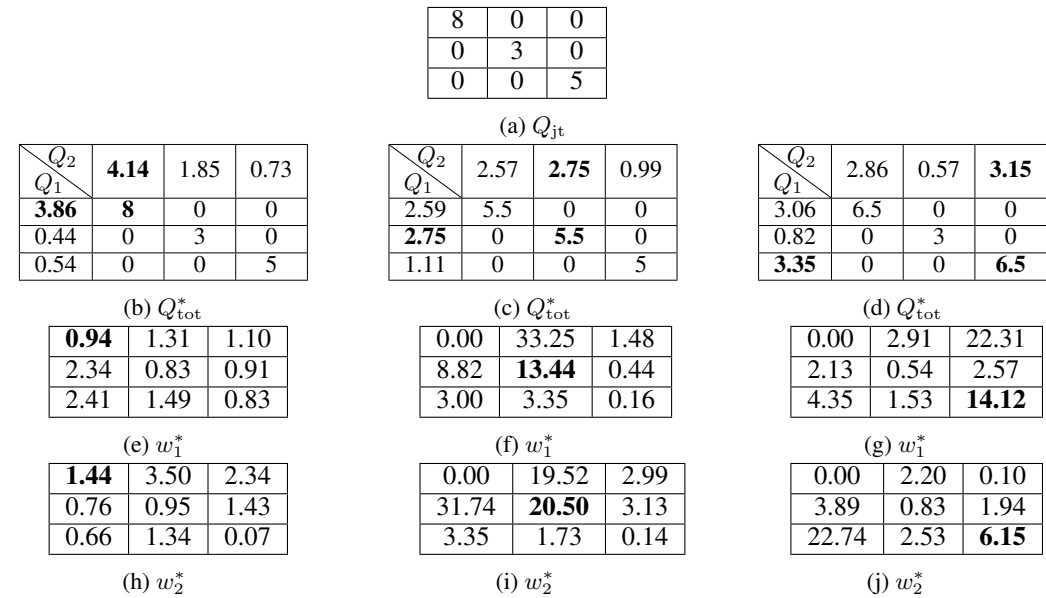

Table 8: The example in which QPLEX has two suboptimal stable points, while QMIX can easily obtain the optimum. The quantities of $Q_{\text{tot}}^*$ and $w_i^*$ listed in the same column correspond to a particular stable point, where we highlight $\overline{\boldsymbol{u}}^*$ in bold.

## E EXPERIMENTAL SETUP

We adopt a setup similar to PyMARL (Whiteson et al., 2019). We use the implementation of QMIX, QTRAN, QPLEX, WQMIX, ResQ, and NA$^2$Q (Liu et al., 2023) from their open-source repositories [5][6][7][8]. These codes are released under the Apache License V2.0.

All algorithms apply TD(0) to update $Q_{\text{tot}}$ including ResQ, as TD($\lambda$) result in catastrophic performance in *5m_vs_6m* while no measurable improvement in other scenarios. The hyperparameters for all algorithms and environments are presented in Table 9. For WQMIX, we set $\alpha = 0.1$ for all environments. For MRVF, we use three rounds of value factorization, with a probability $p = 0.2$ for sampling preselected actions.

For hardware, we run experiments on an NVIDIA 3090 GPU for risk-reward games, predator-prey tasks, and SMAC scenarios (excluding *bane_vs_bane*). In addition, we use an NVIDIA A800 GPU for MRVF in *2c_vs_64zg* and *MMM2* scenarios.

| Hyperparameter | Value | Description |
|---|---|---|
| Batch Size | 32 | Number of episodes per update |
| Replay buffer size | 5000 | Maximum number of stored episodes |
| Target update interval | 200 | Frequency of updating the target network |
| Initial $\epsilon$ | 1.0 | The initial $\epsilon$ in the $\epsilon$-greedy policy |
| Final $\epsilon$ | 0.05 | The final $\epsilon$ in the $\epsilon$-greedy policy |
| Anneal steps for $\epsilon$ | 50,000 | Number of steps for linearly decay of $\epsilon$ |
| Discount $\gamma$ | 0.99 | Discount of future return |
| Test interval | 10,000 | Frequency of test evaluation |
| Test episodes | 32 | Number pf episodes to test |

Table 9: The hyperparameters for experiments

**Network structure** We provide details of the network structures presented in Figure 4. The individual Q network and the mixing network ($Q_{\text{tot}}$) are shown in Figure 9, and the joint action value network ($\hat{Q}_{\text{jt}}$) is shown in Figure 10.

The individual Q network processes the observation using a linear layer followed by a GRU (Chung et al., 2014) with 64 hidden dimensions. The mixing network processes the state with 2-layer linear network (64 hidden dimension) and processes the individual Qs with 2-layer linear network, where the weights and bias (generated by the state's linear processor) have 32 hidden dimension. Inspired by the embedding module (Bengio et al., 2003) (Mikolov et al., 2013), we generate the features of $\bar{u}^k$ by selecting them from the continuous features (128 dimensions) of the observation or state. For the concatenated feature, we double the hidden dimensions to 128.

The network structure of $\hat{Q}_{\text{jt}}$ is similar to that of $Q_{\text{tot}}$, with two key differences: First, we use a 3-layer linear network to process the state. Second, the output of the individual network has 64 dimensions. We input the state to $Q_{\text{jt}}$ in all SMAC scenarios except *3s5z* and *3s_vs_5z*, because masking observations of dead units loses critical information for return prediction in complex scenarios. In the *bane_vs_bane* scenario, we replace the embedding feature with the action index to avoid GPU out-of-memory errors.

---

[5]https://github.com/oxwhirl/pymarl

[6]https://github.com/oxwhirl/wqmix

[7]https://github.com/xmu-rl-3dv/ResQ

[8]https://github.com/zichuan-liu/NA2Q

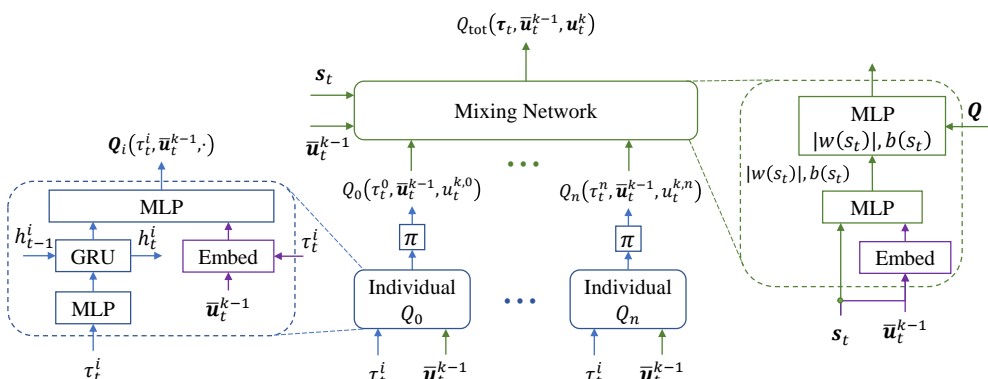

Figure 9: The network structure of $Q_{\text{tot}}$.

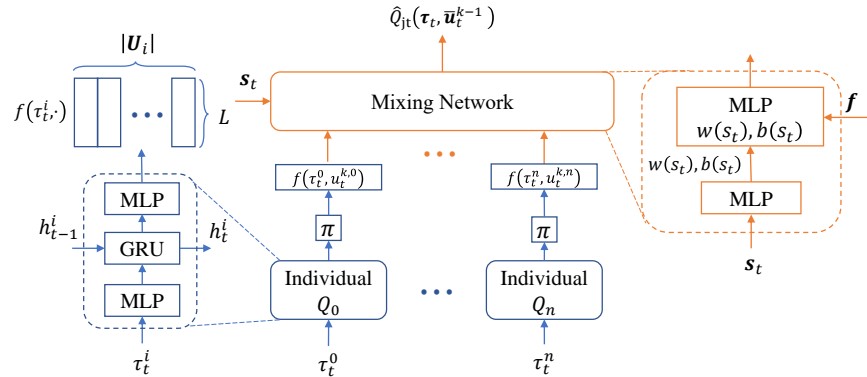

Figure 10: The network structure of $Q_{\text{jt}}$.

### E.1 ONE-STEP GAME

We randomly generate $Q_{\text{jt}}$ of the risk-reward game with the following steps: First, randomly generate the individual reward vectors $r_i \geq 0$. Second, we randomly generate a bijection $\mathcal{U} \to \mathcal{U}$ that maps $u_i$ to another action $v_i$ for all agents $i$, which is the exchange of rows and columns for a matrix. Finally, we let $Q_{\text{jt}}(\boldsymbol{u}) = sign(\boldsymbol{v}) * \sum_{i=0}^{n} r_i(v_i)$ where $sign(\boldsymbol{v}) = 1$ if all agents choose the same mapped action, and $sign(\boldsymbol{v}) = -1$ otherwise. Here, we give an example of the risk-reward matrix in Table 10.

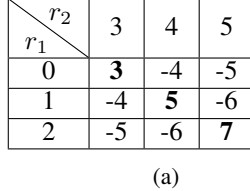

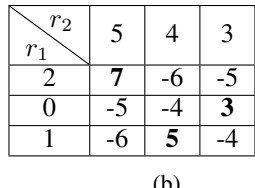

| $r_1$ \ $r_2$ | 3 | 4 | 5 |
|---|---|---|---|
| 0 | **3** | -4 | -5 |
| 1 | -4 | **5** | -6 |
| 2 | -5 | -6 | **7** |

(a)

| $r_1$ \ $r_2$ | 5 | 4 | 3 |
|---|---|---|---|
| 2 | **7** | -6 | -5 |
| 0 | -5 | -4 | **3** |
| 1 | -6 | **5** | -4 |

(b)

Table 10: An example of the risk-reward matrix. (a): the risk-reward matrix before action mapping, where positive rewards are in the diagonal. (b): the risk-reward matrix after action mapping, which scatters positive rewards.

We constrain the values of $Q_{\text{jt}}$ to the interval $[-10, 10]$. To achieve this, we first generate the reward vectors uniformly from $[0, 1]$, then divide each component by $n$, and finally multiply the result by

10. The normalized return in Figure 5 is defined as

$$
return(\boldsymbol{u}) = \begin{cases} \dfrac{Q_{\mathrm{jt}}(\boldsymbol{u}) - \min\limits_{\boldsymbol{u}^+ \in \boldsymbol{\mathcal{U}}^+} Q_{\mathrm{jt}}(\boldsymbol{u}^+)}{\max\limits_{\boldsymbol{u}^+ \in \boldsymbol{\mathcal{U}}^+} Q_{\mathrm{jt}}(\boldsymbol{u}^+) - \min\limits_{\boldsymbol{u}^+ \in \boldsymbol{\mathcal{U}}^+} Q_{\mathrm{jt}}(\boldsymbol{u}^+)} & Q_{\mathrm{jt}}(\boldsymbol{u}) \geq 0 \\[4mm] \dfrac{Q_{\mathrm{jt}}(\boldsymbol{u}) - \min\limits_{\boldsymbol{u}^- \in \boldsymbol{\mathcal{U}}^-} Q_{\mathrm{jt}}(\boldsymbol{u}^-)}{\min\limits_{\boldsymbol{u}^- \in \boldsymbol{\mathcal{U}}^-} Q_{\mathrm{jt}}(\boldsymbol{u}^-)} & Q_{\mathrm{jt}}(\boldsymbol{u}) < 0 \end{cases}
\tag{30}
$$

where $\boldsymbol{\mathcal{U}}^+ = \{\boldsymbol{u}^+ | \boldsymbol{u}^+ \in \boldsymbol{\mathcal{U}}, Q_{\mathrm{jt}}(\boldsymbol{u}^+) \geq 0\}$ and $\boldsymbol{\mathcal{U}}^+ \cup \boldsymbol{\mathcal{U}}^- = \boldsymbol{\mathcal{U}}$.

### E.2 PREDATOR PREY

The predator-prey task (Böhmer et al., 2020) involves a multi-agent scenario where 8 predators cooperate to capture 8 prey. Each predator can choose from six possible actions: four directional movements (up, down, left, right), staying still, or attempting to "capture" prey.

**Capture Conditions**

- A predator can only select the "capture" action when occupying the same position with prey.
- Successful capture requires at least two predators simultaneously choosing "capture" on the same prey.
- Successful capture yields $+10$ reward.
- If only one predator chooses "capture" for a prey, the team receives a punishment.
- The captured prey and successful predators are immediately removed from the environment. Observations of removed agents are masked with zeros.

### E.3 STARCRAFT II MULTI-AGENT CHALLENGE

The SMAC (SC2.4.10 version) involves two opposing teams competing to defeat each other. Players control one team's units with the objective of eliminating all enemy units. The reward system consists of three components:

**Reward**

- **Kill Reward**: $+10$ for eliminating an enemy unit by reducing its HP to zero.
- **Win Reward**: $+200$ for defeating all enemy units and winning the game.
- **Damage**: The amount of change in an enemy's HP.

In the original PyMARL implementation (Whiteson et al., 2019), rewards are enforced as positive by taking absolute values. However, in scenarios where enemies can regenerate health/shields, agents might artificially inflate rewards by allowing recovery and reinflicting damage, rather than focusing on winning. To address this, we adopt the reward without taking its absolute values.

We illustrate why monotonic value factorizations excel in SMAC from two aspects. First, the individual action has slight impact on returns. Take *3s_vs_5z* scenario for example, the enemy unit, Zealot, has 100 health and 50 recoverable shields. Thus, the maximum return is

$$
\underbrace{150*5}_{Damage} + \underbrace{10*5}_{Kill} + \underbrace{200}_{Win} = 1000
\tag{31}
$$

In particular, damages constitutes the major component of return (75% proportion). Within a step, an individual agent's action only affects its damage, and its action will not cause a sharp change of the state. Consequently, a slight deviation from the optimal joint action is insufficient to cause a significant drop in return. Therefore, monotonic value factorization advantages in SMAC, as without the sharp drop in return near the global optimum, monotonic value factorization can attain the global optimum.

Second, choosing the optimal individual action also performs the best in average. We consider a "focus fire" case that frequently appears in SMAC. In this case, two agents fight with three enemies denoted as "A", "B", and "C", and focusing fire on the same target is the trick to win. Those enemies are identical units, except that Enemy A currently has less HP. We present the $Q_{\text{jt}}$ of this case in Table 11. Although $Q_{\text{jt}}$ in Table 11 is non-monotonic, the average return of choosing action A is superior to that of choosing other actions. In this case, monotonic value factorization can obtain the optimal value.

|   | $A$ | $B$ | $C$ | $\emptyset$ |
|---|---|---|---|---|
| $A$ | 2.2 | 1.1 | 1.1 | -0.9 |
| $B$ | 1.1 | 2 | 1 | -1 |
| $C$ | 1.1 | 1 | 2 | -1 |
| $\emptyset$ | -0.9 | -1 | -1 | -2 |

Table 11: $Q_{\text{jt}}$ in the "focus fire" case of the SMAC environment: The action is denoted by its attack target, and the action $\emptyset$ means no attack. The data in the table only represents the relative advantages of each joint action rather than actual values. The agents receive a higher return (+2) when both focus fire on the same target, a moderate return (+1) when attacking different targets, and lower returns for not attacking. Additionally, attacking low-health units (Enemy A) grants a +0.1 bonus to the returns.

# F    ABLATION STUDY

## F.1    EXPERIMENT AND ANALYSIS

We analyze the impact of multi-round iteration and strictness in improving greedy action (Equation (5)). We design MRVF-single, which uses a single round, and MRVF-non-strict, which also uses the original $Q_{\text{jt}}$ as the target value of $Q_{\text{tot}}$ (Equation (6)) for round $k > 1$. We present the results in risk-reward games (Figure 11), predator-prey tasks (Figure 12), and SMAC (Figure 13).

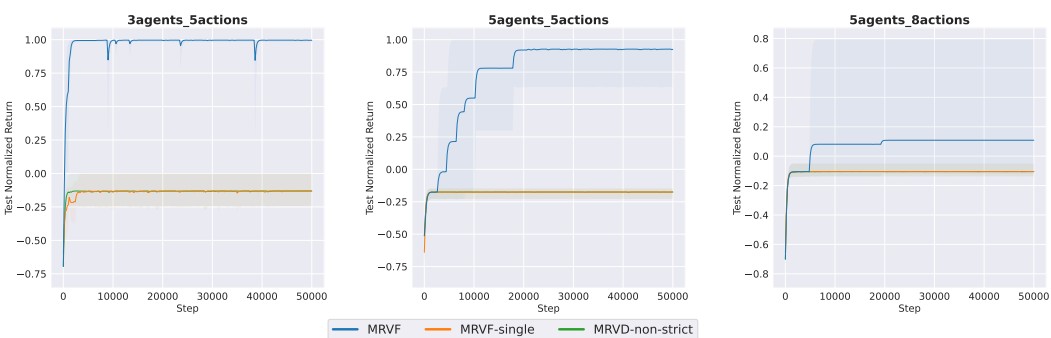

Figure 11: Test normalized return in the risk-reward games.

The results presented in Figure 11 and Figure 12 demonstrate that the strict improvement contributes to better performance in highly non-monotonic scenarios. The explanation for this is that MRVF-non-strict cannot guarantee improvement in the greedy action compared to the preceding round, which prevents it from obtaining the optimal action within a given number of rounds.

As for approximately monotonic scenarios like SMAC, Figure 13 shows a slight improvement compared to single-round factorization. In these scenarios, the improvement comes from additional information in decision-making. Specifically, by feeding $\overline{u}_t^{k-1}$ into the individual Q networks, each agent knows the intentions of others and could infer their states accordingly. Without global information, this additional information contributes to better decision-making. In addition, due to the approximate monotonicity of the payoff, the first round of MRVF already produces a satisfactory solution, which explains the slightness of the improvement.

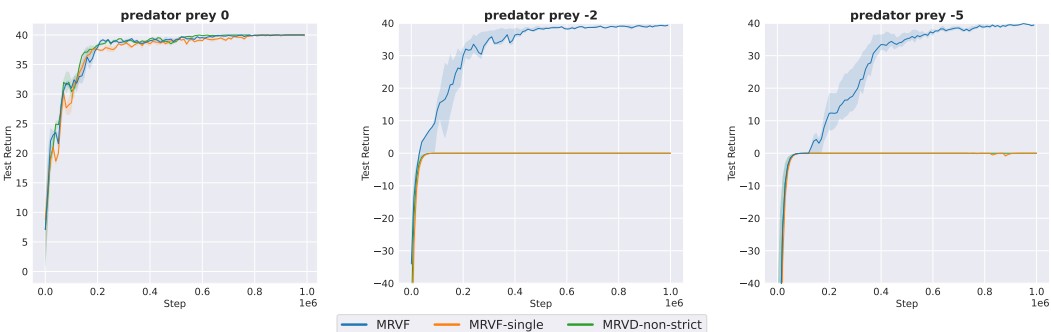

Figure 12: Test return in the predator-prey tasks with punishments 0 (left), -2 (middle), and -5 (right).

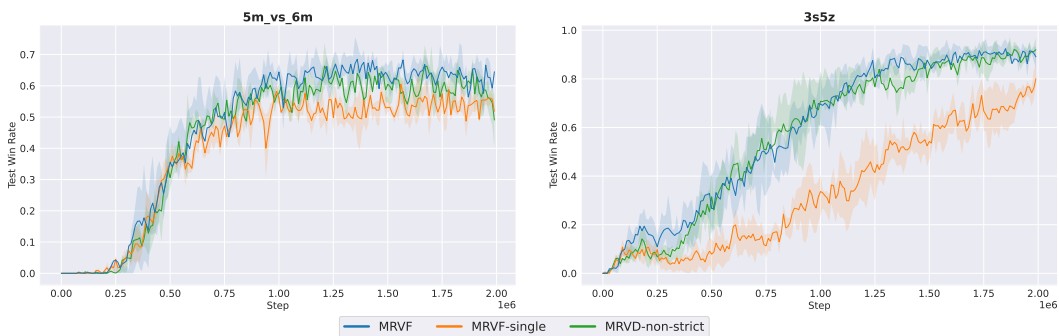

Figure 13: Test win rate in the SMAC benchmarks.

### F.2 DISCUSSION ON ROUNDS

We will discuss how many rounds MRVF requires to obtain the optimal solution under scenarios with different levels of non-monotonicity. As discussed in Section F.1, a second round is required to obtain the optimal solution in highly non-monotonic environments. To support our claim, we calculate the proportion of $\boldsymbol{u}_t^k = \boldsymbol{u}_t$ for each round $k$ throughout test episodes, as shown in Figure 14.

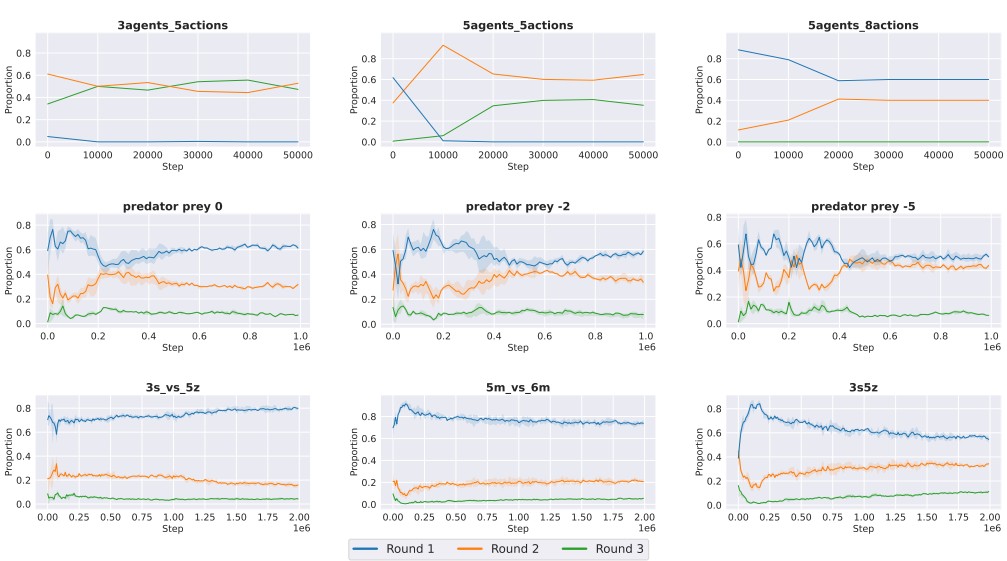

Figure 14: Proportion of final actions generated in each round throughout test episodes.

From Figure 14, we find that in the predator-prey environment, as the penalty increases, the proportion of second-round decisions rises from 30% to nearly 50%. In contrast, for approximately monotonic scenarios such as SMAC, the proportion of second-round decisions typically remains around 20%, and the performance gap between multi-round and single-round is small. In addition, a third round is generally unnecessary unless the reward is so sparse that the second round fails to find the optimal action. As shown in Figure 14, third-round decisions account for only about 5% across all predator-prey and SMAC scenarios. Nevertheless, to balance performance and efficiency, we recommend starting with a maximum of three rounds and adjusting it based on the proportion.

## G    Limitations

Although MRVF achieves significant improvements over existing methods in highly non-monotonic scenarios, it faces the issue of high computational complexity: MRVF requires calculating $Q_{\text{tot}}$ and $Q_{\text{jt}}$ in each round. Although we reduce computational complexity by reusing parts of network outputs (*e.g.*, individual features in $Q_{\text{jt}}$) and early terminating the multi-round iterations for evaluation, the computational complexity of MRVF is still $O(K(N + 1))$, while that of QMIX is at $O(N)$. In addition, as shown in Equation (6) and Equation (5), the update of $Q_{\text{tot}}$ depends on the results of $Q_{\text{jt}}$. As a result, the update of $Q_{\text{tot}}$ is delayed compared to standard TD learning, which leads to a slow rise of the win rate in *3s_vs_5z* (Figure 7). However, learning directly from $Q_{\text{jt}}$ is inevitable for environments that are not retraceable. Otherwise, if exploration of branching trajectories is allowed, TD targets could be used in the update of $Q_{\text{tot}}$.

We summarize the reasons why MRVF may fail to achieve the optimal solution in practice: First, the approximation errors of $\hat{Q}_{\text{jt}}$ propagate to $Q_{\text{tot}}$, which may prevent the $Q_{\text{tot}}$ from reaching the global optimum. Second, monotonic factorization is not ideal. Since we use a neural network as the mixing network, the approximation errors introduced by the network make it difficult to obtain the $Q_{\text{tot}}$ that minimizes $L_{\text{tot}}$. Third, overly sparse rewards may further amplify these approximation errors. As shown in Figure 5, MRVF struggles to achieve optimal performance in *5agents_8actions*.

## H    Supplementary Experiment and Analysis

In this section, we provide analysis and experiments that are not included in the main part of this paper.

### H.1    StarCraft II Multi-Agent Challenge

In Section 3.3, we analyze the stability of QPLEX (Wang et al., 2021), WQMIX (Rashid et al., 2020a) and ResQ (Shen et al., 2022). We reveal that suboptimal stable points are the main cause of the suboptimality in highly non-monotonic scenarios including risk-reward games and predator-prey tasks with large punishments. However, in Figure 7, these methods show inferior performance compared to monotonic value factorizations in SMAC scenarios. We provide the following explanation for this phenomenon.

**QPLEX and ResQ**    From Table 8 in Appendix D, in cases where QMIX can easily obtain the global optimum, QPLEX may still fail due to multiple suboptimal stable points. In addition, for ResQ, Theorem 3 shows that $\boldsymbol{u}^*$ is a weakly stable point in general cases. The weak stability of ResQ is particularly severe because any action can potentially become $\overline{\boldsymbol{u}}$ (Lemma 2). Consequently, QPLEX and ResQ do not perform well in SMAC.

**WQMIX**    WQMIX is designed for highly non-monotonic scenarios, and from Figure 5 and Figure 6, it outperforms monotonic value factorization methods. Although WQMIX is supposed to perform similarly to QMIX in weakly non-monotonic or monotonic scenarios, it shows poor performance in SMAC (Figure 7). We provide three factors resulting in this phenomenon.

(1) **The joint action-value network**: Individual features constitute only a small portion of the mixer's input in the joint action-value network. Consequently, the joint action-value network hardly distinguishes different actions, since action information is primarily embedded in these individual features.

(2) **Decentralized $\epsilon$-greedy policy**: The decentralized $\epsilon$-greedy policy (defined in Equation (2)) combined with the weight in WQMIX enhances the convergence to the local optimum, since the decentralized $\epsilon$-greedy policy explores locally and the weight discards lower values around the local optimum.

(3) **Weight**: When $\alpha < 1$, the weight reduces sample efficiency, as downweighting samples is similar to discarding them.

We conduct additional experiments to support our analysis. We replace the network of $\hat{Q}_{\text{jt}}$ with that of ours and replace the decentralized $\epsilon$-greedy policy with the centralized one. The result in Figure 15 demonstrates that these three factors do result in the performance gap between WQMIX and QMIX.

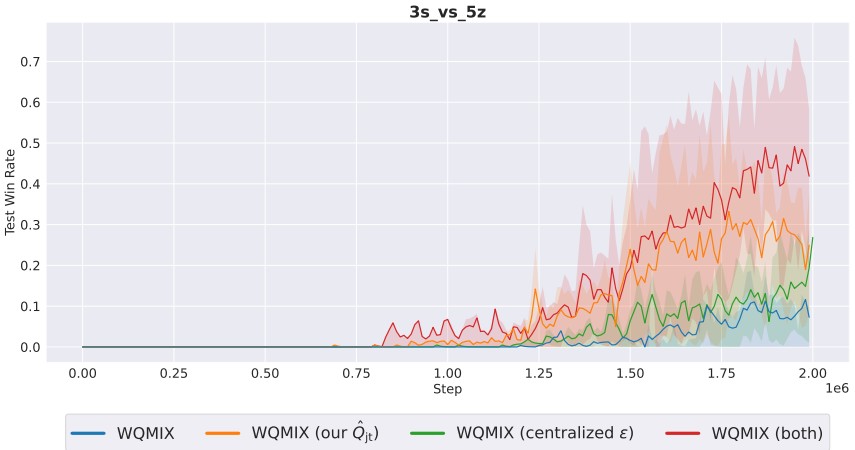

Figure 15: Test win rate in *3s_vs_5z*. WQMIX ($\hat{Q}_{\text{jt}}$) represents WQMIX replacing the network of $\hat{Q}_{\text{jt}}$ with that of ours. WQMIX (centralized $\epsilon$) represents WQMIX replacing the policy with the centralized one. WQMIX (both) represents WQMIX with both modifications.

## H.2 SUPER-HARD SCENARIO OF SMAC

We conduct additional experiments on the *6h_vs_8z* scenario, which is a particularly challenging scenario where the "focus fire" is a key tactic for victory (Whiteson et al., 2019). We adopt a setup similar to the *bane_vs_bane* scenario, except that we set anneal steps for $\epsilon$ to $10^6$. The results in Figure 16 show a slight difference between MRVF and QMIX, supporting our analysis in Appendix E.3.

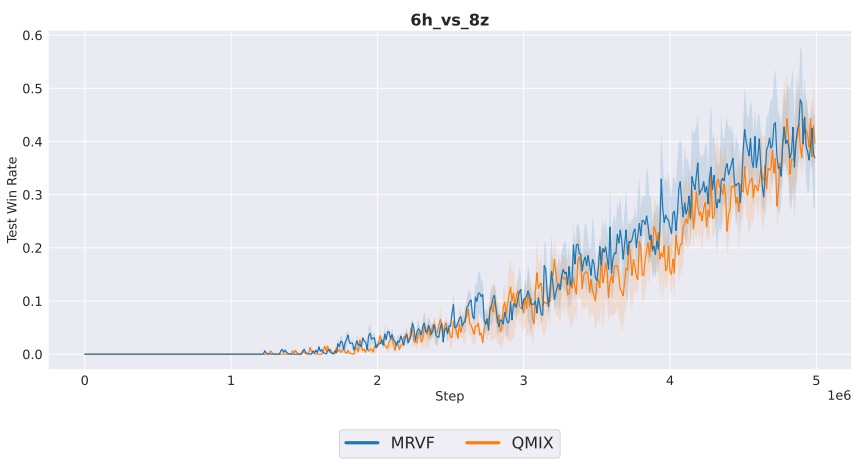

Figure 16: Test win rate in *6h_vs_8z* in SMAC benchmarks.

### H.3 SMACv2

Results on SMACv2 (Ellis et al., 2023) are shown in Figure 17. Overall, the performance gap between MRVF and QMIX is small. This is because SMACv2 inherits the reward design of SMAC; as shown in Appendix E.3, its payoff is nearly monotonic. However, we observe a mismatch between the return and the win rate in SMACv2, suggesting that the reward design may need to account for environmental stochasticity. Due to this mismatch, we present the SMACv2 experiments only in the appendix.

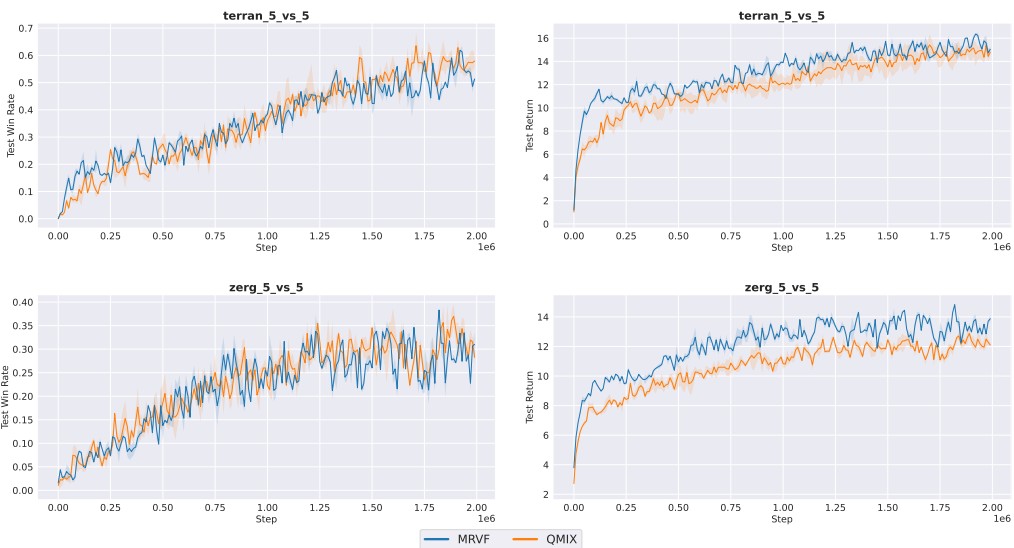

Figure 17: Test win rate (left) and return (right) in SMACv2 benchmarks.

### H.4 PREDATOR PREY

From the results in Figure 6, QMIX fails to obtain the optimal action when the penalty is -2. However, an analysis based on ideal QMIX will reveal that the optimal action is a stable point for any $\epsilon \in (0, 1]$. In fact, this does not indicate a theoretical flaw. An explanation is that suboptimal actions are also stable points as $\epsilon \to 0$.

We will provide an analysis of the stable points when $\epsilon \to 0$. Taking two agents as an example, the reward function for the predator-prey environment with a penalty of -2 is given in Equation (32).

$$reward = \begin{bmatrix} 10 & -2 & \cdots & -2 \\ -2 & 0 & \cdots & 0 \\ \vdots & \vdots & \ddots & \vdots \\ -2 & 0 & \cdots & 0 \end{bmatrix} \tag{32}$$

where the reward is a matrix of shape $6 \times 6$.

Consider $\widetilde{u}$ that neither agent executes the "capture" action, which yields a reward of 0. As $\epsilon \to 0$, $Q^*_{\text{tot}}$ that minimizes $L_{\text{tot}}$ is given in Equation (33).

$$\lim_{\epsilon \to 0} Q^*_{\text{tot}}|_{\widetilde{u} \neq u^*} = \begin{bmatrix} -2 & -2 & \cdots & -2 \\ -2 & 0 & \cdots & 0 \\ \vdots & \vdots & \ddots & \vdots \\ -2 & 0 & \cdots & 0 \end{bmatrix}, L^*_{\text{tot}} = 0 + 0 * o(\epsilon) + 12^2 * o(\epsilon^2) \tag{33}$$

where $\overline{u}^* = \widetilde{u}$, and both correspond to the agents that perform the "non-capture" action. Therefore, "non-capture" action pairs are stable points in predator prey with penalty -2.

Since suboptimal actions are in the majority, it is difficult for the greedy action to converge to the optimal one. However, because suboptimal actions are stable points only when $\epsilon$ is small, we can enhance the convergence to the optimal action by slowing down the decay of $\epsilon$. To support our analysis, we conduct additional experiments on QMIX with $\epsilon$ fixed at 1 and QMIX with $\epsilon$ decaying in one million steps, as shown in Figure 18. From the result in predator prey -2, agents successfully learn to capture prey in both settings, yet they still fail when the penalty is -5. In addition, increasing the weight of the optimal action in the policy can enhance the convergence to the optimal action. For example, the weighting mechanism in WQMIX serves a similar purpose. However, these methods still fail when the penalty is set to -5.

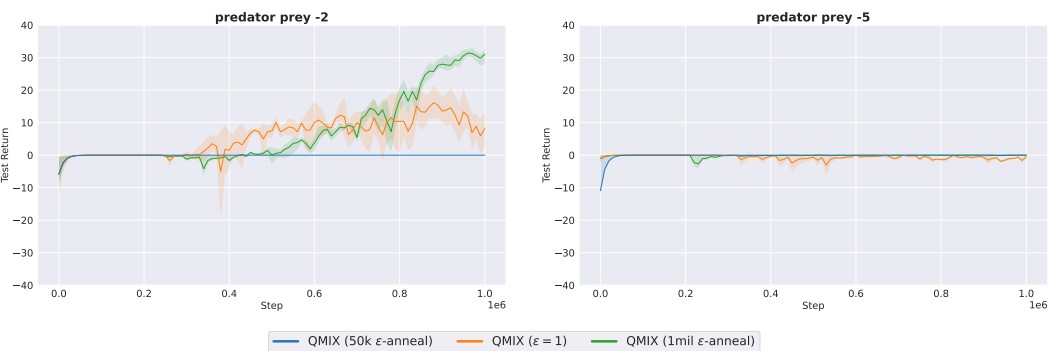

Figure 18: Test return in the predator prey tasks with punishments -2 (left), and -5 (right). QMIX (50k $\epsilon$-anneal) represents QMIX with $5 * 10^4$ anneal steps for $\epsilon$. QMIX (1mil $\epsilon$-anneal) represents QMIX with $10^6$ anneal steps for $\epsilon$. QMIX ($\epsilon = 1$) represents QMIX with $\epsilon = 1$ constantly.

### H.5   COMPARISON WITH OTHER MARL BASELINES

In Section 5, we compare MRVF with algorithms such as WQMIX (Rashid et al., 2020a) that claim to address the representation limitation in value factorization. However, the results show that none of these methods fully resolves this limitation. Other value-factorization approaches, including NA2Q (Liu et al., 2023) and GoMARL (Zang et al., 2023), are not designed to address the representation limitation. On-policy methods such as MAPPO (Yu et al., 2022) suffer from similar issues, converging to suboptimal Nash Equilibria (Ye et al., 2022), which are analogous to suboptimal stable points in value factorization. Consequently, all of these methods perform poorly in environments with highly non-monotonic payoffs, as illustrated in Figure 19.

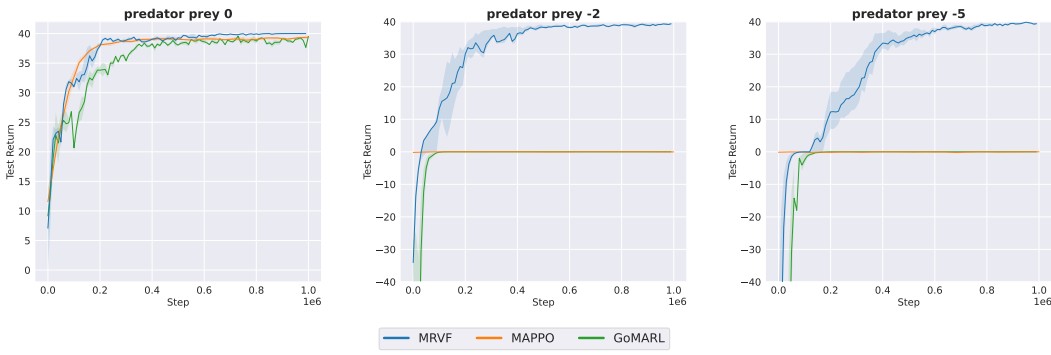

Figure 19: Test return in the predator prey tasks with punishments 0 (left), -2 (middle), and -5 (right).

The comparison results on the SMAC environment are shown in Figure 20. It can be observed that the win rates of the on-policy methods (including MAPPO (Yu et al., 2022) and CommFormer (Hu et al., 2024)) increase more slowly than that of MRVF, and their final performance is usually lower. Moreover, the performance of the on-policy methods depends on hyperparameters, as in Table 12, the network and its inputs for *3s_vs_5z* differ from those used in the other scenarios.

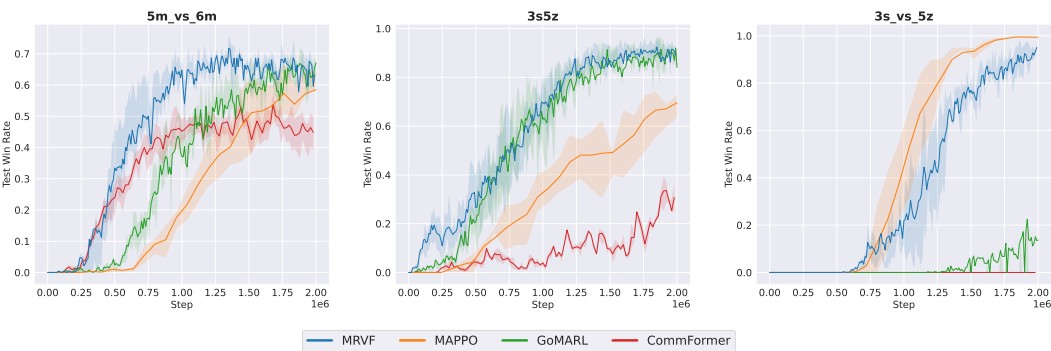

Figure 20: Test win rate in the SMAC benchmarks.

We use the implementation of GoMARL, MAPPO and CommFormer are from their open-source repositories [9] [10] [11]. The hyperparameter settings for the on-policy methods are listed in Table 12. Please refer to (Yu et al., 2022) and (Hu et al., 2024) for further details.

| | epoch | clip | network | stacked frames | training threads | rollout threads |
|---|---|---|---|---|---|---|
| *3s_vs_5z* | 15 | 0.05 | mlp | 4 | 1 | 8 |
| *5m_vs_6m* | 10 | 0.05 | rnn | 1 | 1 | 8 |
| *3s5z* | 5 | 0.2 | rnn | 1 | 1 | 8 |
| predator prey | 10 | 0.2 | rnn | 1 | 1 | 8 |

Table 12: The hyperparameters for on-policy MARL

---

[9]https://github.com/zyfsjycc/GoMARL
[10]https://github.com/marlbenchmark/on-policy
[11]https://github.com/charleshsc/CommFormer

