# OpenReview forum: "MRVF: Multi-Round Value Factorization with Guaranteed Iterative Improvement for Multi-Agent Reinforcement Learning"
_ICLR.cc/2026/Conference — ICLR 2026 Conference Withdrawn Submission_

### Official Review · Reviewer_HHAe · 2025-10-20

**Soundness:** 3
**Presentation:** 3
**Contribution:** 3
**Rating:** 8
**Confidence:** 4

**Summary:**

The paper addresses the optimality of current value factorization methods in multi-agent reinforcement learning (MARL). It formulates a stable point theory for analyzing convergence to a global optimum, i.e., optimal joint actions in a multi-agent system. The paper provides one-shot examples for all common factorization methods, such as VDN, QMIX, WQMIX, and QPLEX, showing that they can fail to converge to the global optimum due to restricted representations or strong assumptions about the greedy joint action of Q-values, which is commonly but misleadingly assumed to be optimal. As a solution, the paper proposes a multi-round value factorization (MRVF) framework, which "smoothes" the value landscape by clipping local optima until only monotonic convergence to the global optimum is ensured. MRVF is evaluated in one-shot games, a predator-prey task, and the StarCraft Multi-Agent Challenge (SMAC), and demonstrated to be more effective than prior value factorization methods.

**Strengths:**

**Novelty**

The paper addresses an important problem with a novel theory and approach. Since I am not aware of similar work, I consider the contributions novel.

**Soundness**

The theory is straightforward, and the counterexamples provided to show the suboptimality of current approaches are sound.

**Quality**

The paper is well-written and easy to follow. The approach and examples are nicely illustrated in figures.

**Significance**

The paper provides an important contribution to the MARL community that can move the field forward. I appreciate the variety in benchmark domains from one-shot to SMAC, which provides sufficient empirical evidence that the approach is effective and scalable.

**Weaknesses:**

SMAC is considered outdated and largely solved, which is also evident in the paper, where MRVF does not dominate the baselines consistently. Furthermore, SMAC is criticized for having deterministic initial states and observations [1,2,3], which has led to more general benchmark suites, such as SMACv2 [2].

**Literature**

[1] Lyu et al., "A Deeper Understanding of State-Based Critics in Multi-Agent Reinforcement Learning", AAAI 2022

[2] Ellis et al., "SMACv2: An Improved Benchmark for Cooperative Multi-Agent Reinforcement Learning", NeurIPS Benchmarks 2023

[3] Phan et al., "Attention-Based Recurrence for Multi-Agent Reinforcement Learning under Stochastic Partial Observability", ICML 2023

**Questions:**

Optimizing over multiple rounds incurs more computational overhead. Do you have data on how this affects the MRVF?

---

> ### Author Response · Authors · 2025-11-15
>
> Thank you for your constructive feedback and acknowledgment of our efforts. Following are our responses to all your concerns.
>
> ## 1. Regarding SMACv2 (W1)
>
> This suggestion is really helpful. We will evaluate MRVF in SMACv2 and update the result in revisions.
>
> ## 2. Regarding runtime (Q1)
>
> We mention in the Limitations (Appendix G) that the complexity of MRVF is $K$ times that of QMIX, where $K$ is the maximum number of rounds. We provide the average runtime per step during training as follows:
>
> **Table: Average runtime (seconds) per step  across different scenarios**
>
> |            | 5m_vs_6m | 3s5z   | bane_vs_bane |
> |-----------|----------|--------|--------------|
> | **MRVF**      | **0.031**    | **0.048**  | **0.094**        |
> | QMIX      | 0.012    | 0.015  | 0.031        |
> | QPLEX     | 0.013    | 0.016  | 0.038        |
> | ResQ      | 0.020    | 0.021  | 0.050        |
>
> In this paper, we use $K=3$. From the Table, the time complexity is approximately three times that of QMIX, which supports our description in the Limitations section.
>
> ## 3. Future work
>
> Here are some of our thoughts on the future work of MRVF, and we are very pleased to share them with you.
>
> We notice that existing value factorization methods perform poorly in environments with coupled action constraints. In such environments, penalties for violating these constraints are typically used to ensure policies are feasible and safe.
>
> For example, in the *predator-prey* environment, two agents must simultaneously choose either "catch" or "not catch"; otherwise, they receive a penalty. Generally, a larger penalty leads to a safer policy. However, as the *predator-prey* results demonstrate, excessively large penalties cause existing algorithms to adopt overly conservative policies, forsaking optimal ones. In contrast, MRVF achieves the optimal policy even with a penalty -5, indicating its superior capability in handling coupled action constraints.
>
> However, we have yet to find a benchmark with both coupled action constraints that has authority comparable to SMAC. It would be great help if you could recommend such a benchmark.

---

### Official Review · Reviewer_RKef · 2025-10-31

**Soundness:** 3
**Presentation:** 3
**Contribution:** 3
**Rating:** 6
**Confidence:** 3

**Summary:**

This paper studies the convergence properties of value-factorization MARL methods and identifies that single-round monotonic factorizations inevitably converge to suboptimal stable points in highly non-monotonic coordination tasks. It introduces MRVF, a multi-round value factorization framework that iteratively refines joint-action estimates by using payoff increments clipped by previous greedy values, guaranteeing strict improvement of the greedy action until the optimal action is reached. The method is evaluated on multiple benchmarks, showing substantial gains in challenging non-monotonic settings and competitive or better results in monotonic ones.

**Strengths:**

1. The paper introduces a formal definition of stable points in MARL value factorization and proves why existing methods get stuck in suboptimal equilibria.
2. The paper reframes MARL training dynamics similar to multi-round communication or iterative refinement.
3. The paper conducts multiple domains and ablations, especially in non-monotonic cases.

**Weaknesses:**

See the questions section.

**Questions:**

1. What is the computational overhead of multiple rounds per timestep in SMAC and larger benchmarks?
2. How sensitive is performance to the number of rounds and clip threshold?
3. How is MRVF compared to other methods, such as MAPPO methods?

---

> ### Author Response · Authors · 2025-11-15
>
> Thank you for your insightful questions and comments. Following are our responses to all your concerns.
>
> ## 1. Regarding Runtime (Q1)
>
> We mention in the Limitations (Appendix G) that the complexity of MRVF is $K$ times that of QMIX, where $K$ is the maximum number of rounds. We provide the average runtime per step during training as follows:
>
> **Table: Average runtime (seconds) per step  across different scenarios**
>
> |            | 5m_vs_6m | 3s5z   | bane_vs_bane |
> |-----------|----------|--------|--------------|
> | **MRVF**      | **0.031**    | **0.048**  | **0.094**        |
> | QMIX      | 0.012    | 0.015  | 0.031        |
> | QPLEX     | 0.013    | 0.016  | 0.038        |
> | ResQ      | 0.020    | 0.021  | 0.050        |
>
> In this paper, we use $K=3$. From the Table, the time complexity is approximately three times that of QMIX, which supports our description in the Limitations section.
>
> ## 2. Sensitivity Analysis (Q2)
>
> ### 2.1 The number of rounds
> As discussed in our Appendix F.2, the impact of the number of rounds on performance depends on the monotonicity of the environment's payoff structure. For highly non-monotonic payoffs, such as in *predator-prey*, at least two rounds are required to achieve the optimal solution. For monotonic payoffs, such as in SMAC, the policy improvements beyond the second round are negligible. In our experiments, we used a three-round MRVF configuration to demonstrate in Appendix F.2 that two rounds are generally sufficient.
>
> ### 2.2 The clip threshold
>
> We do not use a clip threshold. Actually, we attempted to incorporate a threshold for the action-value increment but observed no notable impact on performance. Consequently, we decided to remove this component for better clarity.
>
> ## 3. Comparison to MAPPO (Q3)
>
> We will include comparative experiments with MAPPO in our revision.
>
> However, it should be noted that there are significant differences in the experimental settings between MAPPO (an on-policy method) and MRVF (an off-policy method). For instance, MAPPO utilizes multiple threads for rollout, which results in collecting more samples per step. Therefore, the experimental results are only for reference.

---

### Official Review · Reviewer_iyyU · 2025-11-01

**Soundness:** 3
**Presentation:** 1
**Contribution:** 2
**Rating:** 4
**Confidence:** 4

**Summary:**

This paper introduces Multi-Round Value Factorization (MRVF), a framework for cooperative multi-agent reinforcement learning that reformulates value decomposition as a multi-round optimization process. Instead of minimizing a single static loss as in WQMIX or QPLEX, MRVF uses the greedy joint action from the previous round as a dynamic baseline and fits only the positive increments in joint-action value, enforcing a strict improvement condition. The authors prove that, under finite action spaces, this procedure guarantees convergence to the globally optimal joint action. Empirical examples illustrate that MRVF avoids suboptimal stable points observed in prior factorization methods.

**Strengths:**

- The paper provides a theoretical perspective on why existing value factorization methods may converge to suboptimal stable points.

- The proposed multi-round optimization view and strict-improvement condition is sound.

**Weaknesses:**

- Presentation and notation are not clear. The use of $u$, $\tilde{u}$, and $\bar{u}$ is confusing and imprecise, making the paper hard to follow for readers not already familiar with value-decomposition methods such as WQMIX.

- Related work is incomplete. Many issues discussed such as suboptimal convergence and stability in MARL have been studied before [1], so the contribution appears incremental.

- The difference from WQMIX ($\alpha=0$) and QTRAN is unclear. There is no sufficient analysis or ablation explaining why MRVF performs better on SMAC.


[1] Ye, Jianing, Chenghao Li, Jianhao Wang and Chongjie Zhang. “Towards Global Optimality in Cooperative MARL with the Transformation And Distillation Framework.” (2022).

**Questions:**

1. The presentation should be rewritten for clarity. Many notations and terms (e.g., the several variants of $u$, as well as the definition of “value decomposition” itself) are introduced informally and used inconsistently. Please make these concepts explicit and mathematically precise.

2. What is the concrete difference between the proposed multi-round optimization and simply setting $\alpha = 0$ in WQMIX?

3. If WQMIX uses $\alpha = 0$, or equivalently if we use QTRAN (which can be viewed as a weighted VDN with $\alpha = 0$), wouldn’t suboptimality problem disappear entirely? In that case, any non-optimal greedy action would yield a positive loss, and only when the greedy action is the true $\arg\max Q_{jt}$ would the loss be zero (since other entries are masked out). How does the proposed MRVF differ from this behavior?

4. (Minor) The paper claims that QPLEX fails because suboptimal actions are not unstable. However, QPLEX explicitly introduces a stop-gradient trick in its loss to ensure that suboptimal actions do have nonzero gradients, thus avoiding exactly this issue. How does the proposed analysis reconcile with that fact?

---

> ### Author Response · Authors · 2025-11-14
>
> Thank you for your helpful comments and suggestions. We would like to provide clarification regarding your concerns:
>
> ## 1. Notations and Terms (W1, Q1)
> ### 1.1. The several variants of $u$
> The several variants of  $u$ are detailed in the background (mainly in Section 2.3), along with many examples. Additionally, in case readers skip the background, we reiterate their distinctions at the beginning of Section 3.
>
> The difference between the two is subtle, thus treating both as greedy actions does not affect the understanding of the main body of the paper. Moreover, after reading the main part, readers will find that $\widetilde{u} = \bar{u}^k$, while $\bar{u} = \bar{u}^{k+1}$. However, we cannot define them this way because the concept of multiple rounds was not present in prior work.
>
> ### 1.2. "Decomposition" or "Factorization"
> Although their meanings are similar, we believe that "Value Factorization" is more precise in this paper:
>
> First, "Factorization" is used instead of "Decomposition" in QMIX [1], QTRAN [2], QPLEX [3], WQMIX [4], and ResQ [5]. Among these, QMIX is a classic and influential work on value factorization.
>
> Second, in QTRAN and WQMIX, "Decomposition" specifically refers to the additive form. Therefore, it is also reasonable for NA²Q [6] to use "Decomposition," as it employs Taylor expansion.
>
> In summary, "Factorization" is more general, while "Decomposition" specifically refers to the additive form of factorization. Since MRVF uses a network as a mixer, "Factorization" is more precise.
>
> ### 1.3. Suggestions for readers unfamiliar with value factorization
> Understanding the theoretical section of this paper requires background knowledge. Readers without it are advised to skip this part, as it does not affect the comprehension or application of the MRVF method. For those wishing to go further, we advise returning to the theoretical section, as it provides guidance for the design of factorization.
>
> We thank the reviewer for raising this important issue. Since this comment is public, we place these suggestions here to express our appreciation.
>
> ## 2. Regarding the paper by Ye et al. [7] (W2, Q4)
> First, discussing the suboptimality of existing work is quite common in MARL. Therefore, we only include highly relevant studies in our references, such as GVR [8], because the self-transition node (STN) in this work can be viewed as a specific form of stable point.
>
> Second, we find some issues in the paper by Ye et al. [7]: QPLEX is rewritten as:
> $$Q_{tot}=b(s;\psi)+\sum\lambda_i(s,a;\phi)A_i(s,a;\theta)$$
> However, in the original version of QPLEX, $b(s;\psi)$ should be $\sum max Q_i$. In addition, Ye et al. find $\psi, \phi$ by keeping $\theta$ fixed, which is confusing since $\phi$ and $\theta$ should be updated synchronously. This issue is also raised in OpenReview for ICLR 2023. Finally, the conclusion is inconsistent with the results in Table 7 (c) and Table 8 (c) from Appendix D of our paper.
>
> In summary, we do not find the similarity between "stable point" and the proof by Ye et al., and it seems that the proof by Ye et al. is somewhat problematic.
>
> Regarding the stop-gradient trick (Q4), we do not find it in either the paper or the code of QPLEX. We note that the expression "In empirical design" is used in [7], which might suggest that the trick is originally proposed. Therefore, it would be helpful if more details are provided.
>
> ## 3. About WQMIX and MRVF (W3, Q2, Q3)
> First, $\alpha$ cannot be that small, which is discussed in both [4] and Appendix H.1 of our paper. Even if $\alpha=0$, $u^* $ is a **weak** stable point (as shown in Table 1).
>
> Second, the action-value increment can be replaced with other forms that satisfy the "Strict Improvement Condition". However, since the action-value increment fails to satisfy the "Strict Improvement Condition" when $u^* $ is found, multiple rounds are required to record previous results and enable early termination. We hope that future work will introduce better designs for this part.
>
> ## Literature
>
> [1] Rashid et al., ". Monotonic value function factorisation for deep multi-agent reinforcement learning", JMLR 2020
>
> [2]  Son et al., "Qtran: Learning to factorize with transformation for cooperative multi-agent reinforcement learning", ICML 2019
>
> [3] Wang et al., "Qplex: Duplex dueling multi-agent q-learning", ICLR 2021
>
> [4] Rashid et al., "Weighted qmix: expanding monotonic value function factorisation for deep multi-agent reinforcement learning", NeurIPS 2020
>
> [5] Shen et al., "Resq: A residual q function-based approach for multi-agent reinforcement learning value factorization", NeurIPS 2022
>
> [6] Liu et al., "NA²Q: Neural attention additive model for inter-pretable multi-agent q-learning", ICML 2023
>
> [7] Ye et al., "Towards Global Optimality in Cooperative MARL with the Transformation And Distillation Framework", 2022
>
> [8] Wan et al., "Greedy based value representation for optimal coordination in multi-agent reinforcement learning", ICML 2022

---

> ### Author Response · Authors · 2025-12-02
> **Regarding the paper by Ye et al**
>
> Ye et al. ignore the impact of changes in $\bar{u}$ on the gradient, leading to a severe overestimation of suboptima. In contrast, our analysis based on stable points addresses this issue.
>
> The main reason for this overestimation is that, in Lemma B.13 of the latest arXiv version (2023) of [1], the authors ignore the situation that two individual Q-values in $S=argmin_{f \in R(a^{\*})}{L(f)}$  may be equal. Whenever this occurs, there exist two distinct $a^{\*}$  (corresponding to  $\bar{u}$ in our paper) in the neighbourhood of the individual Qs. Ye et al. assume only one $a^{\*}$ exists in that neighbourhood, and incorrectly conclude that a zero-gradient point exists for every $a^{\*}∈A$ (denoted  $\tilde{u}$ in our paper).
>
> The stable point we propose handles this situation: when  $\tilde{u}$ is not the optimal action, Equation (28) in our paper may fail to hold under the condition  $\bar{u}=\tilde{u}$.
>
> [1] Ye et al., "Towards Global Optimality in Cooperative MARL with the Transformation And Distillation Framework", 2022

---

### Official Review · Reviewer_59Yi · 2025-11-01

**Soundness:** 2
**Presentation:** 3
**Contribution:** 3
**Rating:** 4
**Confidence:** 4

**Summary:**

The paper proposes MRVF, a framework for cooperative MARL that iteratively monotonizes a non-monotonic joint payoff landscape by training with non-negative payoff increments relative to the previous round’s greedy action.

**Strengths:**

- This paper is generally well written and easy to follow.

- The stable point lens to analyze greedy-action convergence for value factorization is insightful. This paper shows that WQMIX can prefer suboptimal stable points and QPLEX/ResQ have their own stability failure modes.

- This theoretical contribution seems correct and meaningful, with multi-round clipping target simple yet effective for preventing stable points under idealized mixer.

**Weaknesses:**

- I'm mostly concerned about the experimental setup. The SMAC is relatively considered outdated, given it's almost 2026 and that there is already SMAC2, some of the baseline selection are also outdated, with the latest being NA2Q, MAPPO and other more recent baselines are missing.

- Also it seems the experimental results are somewhat different from past works, e.g. in NA2Q most of the baselines could reach 60%+ at 1e6 steps while in this papers result, none of them could reach 40% win rate, I wonder if the experimental setup are different? More details on this would help.

**Questions:**

Please see weakness

---

> ### Author Response · Authors · 2025-11-14
>
> Thank you for your helpful comments and suggestions. We would like to provide clarification regarding your concerns:
>
> ## 1. Regarding SMACv2 (W1)
> This suggestion is really helpful. We will evaluate MRVF in SMACv2 and update the result in revisions.
>
> ## 2. Regarding baseline selection (W1)
> First, apart from classic works in value factorization such as QMIX, the baselines we selected are primarily within recent five years. These works are contemporaneous with MAPPO [1] (2022). Therefore, we believe our baseline selection can hardly be considered outdated.
>
> Second, the field of MARL comprises many specialized subfields, generally categorized into online and offline learning, with online learning further divided into on-policy and off-policy methods. Different subfields often employ distinct experimental setups that are generally incomparable. For instance, the baselines in DoF [2] (an off-policy method) does not include any value factorization algorithms.
>
> **Finally, we will include comparative experiments with MAPPO in our revision.** Regarding more recent baselines: prior to our proposal of the "stable point" concept, the root causes of suboptimality in value factorization algorithms remained unclear, which hindered progress in this research direction. Consequently, it has been challenging to find relevant algorithms published in 2024. It would be helpful if you could find recent value factorization algorithms, and we would like to incorporate them into our experiments. Additionally, **if you could provide a specific list of other MARL baselines** from your subfield that are suitable for comparison, we would be happy to include them in our experiments as well.
>
>
> ## 3. Regarding experiment results (W2)
> The description in the W2 appears to exaggerate the differences: The results for *5m_vs_6m* and *3s5z* are consistent with those reported in NA²Q [3], while *bane_vs_bane* is not evaluated in NA²Q. For *MMM2*, our reported results are even higher than those in NA²Q. In summary, the differences are primarily in *3s_vs_5z* and *2c_vs_64zg*.
>
> ### 3.1. Regarding *3s_vs_5z*
> We find and fix an issue in the reward setup within PyMARL, as detailed in Appendix E.3 (lines 1173-1176). This issue is particularly noticeable in the *3s_vs_5z* scenario because Zealots cannot catch up with Stalkers, allowing Stalkers to continuously deal damage and accumulate rewards until exceeding the step limit. However, this problem had not been reported before. We find it while training MRVF, as MRVF achieved higher returns than all baselines but exhibited a significantly lower win rate. This result strongly demonstrates MRVF's capability of finding optimal solutions (solutions with the highest return).
>
> ### 3.2. Regarding *2c_vs_64zg*
> Actually, we are puzzled by the results reported in NA²Q: the baseline performance it reports for *2c_vs_64zg* surpasses that reported by existing works [4,5,6], even significantly exceeding the self-reported performance from the baseline paper [5]. However, we are unable to replicate these results with the same setup.
>
> ## Literature
> [1] Yu et al., "The Surprising Effectiveness of PPO in Cooperative Multi-Agent Game", NeurIPS 2022
>
> [2] Li et al., "DoF: A Diffusion Factorization Framework for Offline Multi-Agent Reinforcement Learning", ICLR 2025
>
> [3] Liu et al., "NA²Q: Neural attention additive model for inter-pretable multi-agent q-learning", ICML 2023
>
> [4] Wang et al., "Qplex: Duplex dueling multi-agent q-learning", ICLR 2021
>
> [5] Yang et al., "Qatten: A General Framework for Cooperative Multiagent Reinforcement Learning", CoRR 2020
>
> [6] Shen et al., "Resq: A residual q function-based approach for multi-agent reinforcement learning value factorization", NeurIPS 2022

---

> > ### Comment · Reviewer_59Yi · 2025-11-20
> >
> > I thank the authors for their response, this has addressed some of my concerns, I have adjusted my rating accordingly

---

### Author Response · Authors · 2025-12-03
**Summary of Revision**

Our revision has now been updated. The revision primarily addresses the common concerns raised by the reviewers:

1. Added experimental results on the SMACv2 environment in Appendix H.3.

2. Added comparison with additional baselines, including MAPPO [1], GoMARL [2], and Commformer [3], in Appendix H.5.

[1] Yu et al., "The Surprising Effectiveness of PPO in Cooperative Multi-Agent Game", NeurIPS 2022

[2] Zang et al., "Automatic grouping for efficient cooperative multi-agent reinforcement learning", NeurIPS 2023

[3] Hu et al., " Learning multi-agent communication from graph modeling perspective.", ICLR 2024

---

### Note · Authors · 2026-01-09

I have read and agree with the venue's withdrawal policy on behalf of myself and my co-authors.